# MULTI-TOKEN POLICY GRADIENT OPTIMIZATION

## ABSTRACT

Policy-gradient optimization methods like PPO typically operate at the token level, estimating action probabilities for each next-token prediction. While effective, this formulation overlooks the structured nature of reasoning, where meaningful decisions often span multiple tokens—such as defining variables or composing equations. To bridge this gap, we propose **M**ulti-token **P**olicy Gradient **O**ptimization (**MPO**), a framework that treats contiguous blocks of $K$ tokens as unified semantic actions. This block-level perspective better captures the compositional structure of reasoning trajectories and supports optimization over coherent, higher-level objectives. Experiments on mathematical reasoning and coding benchmarks show that MPO consistently outperforms standard token-level policy gradient baselines, demonstrating the effectiveness of modeling multi-token actions for structured reasoning in LLM post-training.

## 1 INTRODUCTION

Large language models (LLMs) have become the foundation of modern natural language understanding and generation, achieving remarkable results through large-scale pretraining and autoregressive modeling (Kumar, 2024; Zhu et al., 2024; Tang et al., 2025). Recently, there has been increasing interest in leveraging policy gradient methods to further fine-tune these models, with the aim of enhancing their capacity for complex reasoning, long-horizon dependency modeling, and alignment with human-preferred behaviors by directly optimizing sequence-level objectives (Ouyang et al., 2022). Among these methods, Proximal Policy Optimization (PPO) has emerged as a dominant framework, offering efficient and stable updates for reinforcement learning-based fine-tuning in LLMs (Schulman et al., 2017). Building on PPO, approaches such as VinePPO introduce intermediate rollouts to facilitate more accurate long-context credit assignment, thereby enhancing policy optimization in autoregressive frameworks (Kazemnejad et al., 2025). Other advanced methods, such as Group Relative Policy Optimization (GRPO) (Shao et al., 2024) and Decoupled Clip and Dynamic Sampling Policy Optimization (DAPO) (Yu et al., 2025), adopt group-based and adaptive sampling strategies to better address the challenges of structured reasoning and mathematical problem solving. These methods compute advantages and manage learning signals over groups of samples, further improving the optimization process.

Although policy gradient techniques have enhanced the reasoning and sequence modeling abilities of LLMs, they remain fundamentally constrained by the next-token prediction (NTP) paradigm (Bachmann & Nagarajan, 2024). As shown in Figure 1 left, in tasks such as mathematical reasoning, decision-making evolves over semantic units—for example, defining variables or completing equations—rather than individual tokens. However, next-token objectives decompose these structured reasoning steps into a series of local token predictions, disrupting the coherence of the underlying semantic actions (Figure 1 right). This mismatch between the token-level optimization granularity and the structured nature of reasoning leads to myopic behaviors: the optimization process focuses on locally plausible token actions instead of considering the integrity of higher-level reasoning progressions (Mirzadeh et al., 2024; Zhang et al., 2025). Consequently, it struggles to sustain consistent symbolic structures or multi-step dependencies. Overcoming this structural fragmentation calls for training strategies that capture reasoning actions at a coarser, semantically meaningful level beyond single-token updates.

To deal with the limitation, we introduce *Multi-token Policy Gradient Optimization* (MPO), a novel framework that incorporates multi-token, block-level actions into the policy gradient process. As illustrated in Figure 1 right, instead of treating each token generation as an isolated action, MPO

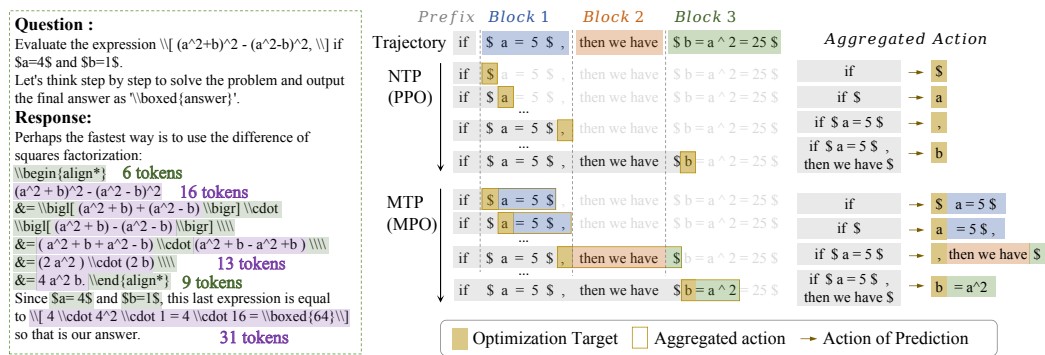

Figure 1: **Left**: In reasoning tasks such as mathematical problem-solving or code generation, the model's decision process often spans across blocks of tokens—such as equations or functions—rather than being determined by each token independently; **Right**: illustration of token-level (NTP/PPO) vs. block-level (MTP/MPO) optimization. MPO aggregates future K actions as a semantically meaningful block for prediction and optimization, thereby better capturing sequence structure and long-range dependencies.

aggregates contiguous blocks of $K$ future tokens and unites importance sampling ratios over them. MPO allows the model to consider multiple correlated tokens as a single semantic action, better preserving the internal structure of reasoning steps such as variable definitions, function calls, or equation formations. This block-level optimization encourages the policy to plan over meaningful reasoning segments rather than isolated symbols, thus maintaining consistency across intermediate decisions and improving global reasoning coherence. Because MPO only modifies the importance sampling ratio, it remains broadly compatible with existing policy-gradient frameworks and can be seamlessly integrated into contemporary LLM post-training pipelines. Our main contributions are:

1. We propose **Multi-token Policy Gradient Optimization (MPO)**, which generalizes token-level policy optimization by treating contiguous blocks of $K$ tokens as unified semantic actions. This design enables the policy to optimize over structured reasoning units rather than isolated symbols.

2. We are the first to incorporate structural multi-token optimization into the post-training stage of large language models, revealing its potential to enhance reasoning coherence and informing new directions for policy-gradient research.

3. We empirically validate MPO on mathematical reasoning benchmarks GSM8K and MATH and coding generation benchmark HumanEval, where it consistently outperforms standard token-level policy gradient baselines, demonstrating its effectiveness in improving structured reasoning ability during post-training.

## 2 RELATED WORK

### 2.1 MULTI-TOKEN PREDICTION WITH LANGUAGE MODELS

Multi-token prediction (MTP) is an extension of standard auto-regressive language modeling, where typically, the model is trained to predict only the immediate next token given the preceding context. In MTP, the model is trained to predict multiple subsequent tokens in a single forward process. Gloeckle et al. (Gloeckle et al., 2024) propose using multiple prediction heads to forecast several future tokens from each context position, improving sample efficiency and inference speed on coding and generative benchmarks. This approach demonstrates stronger induction and reasoning ability gains, particularly at larger model scales like DeepSeek V3 (Liu et al., 2024). Gerontopoulos et al. (Gerontopoulos et al., 2025) extend this to freeze pre-trained models and add learnable "register tokens" to support multi-token prediction without modifying the backbone. While previous works

on MTP enhanced LLM's sample efficiency and accelerated inference, our work extends the MTP concept from pre-training or inference use into the realm of policy-gradient optimization.

## 2.2 POST-TRAINING RLHF ALGORITHMS

Reinforcement Learning from Human Feedback (RLHF) is an approach to align language models with human preferences by optimizing the policy using reinforcement learning on preference-labeled data. Within RLHF, policy-gradient methods have advanced from the basic Proximal Policy Optimization (PPO) to more sophisticated variants designed specifically for large language models (LLMs). PPO (Schulman et al., 2017) introduced the clipped surrogate objective to stabilize updates, becoming the backbone of early RLHF such as InstructGPT (Ouyang et al., 2022). Group Relative Policy Optimization (GRPO) (Shao et al., 2024), built on PPO, computes advantages by comparing multiple sampled completions per prompt (group-based advantage), eliminating the need for a value network and enhancing learning on reasoning tasks. DAPO (Decoupled Clip and Dynamic Sampling Policy Optimization) (Yu et al., 2025) further refines GRPO's token-level loss by allowing asymmetric clipping bounds ("clip-higher") and dynamic sampling to maintain helpful gradient signal. VAPO (Yue et al., 2025) integrates value-based methods into reasoning and RLHF, augmenting PPO variants with adaptive GAE and value pretraining for enhanced stability and performance. CISPO (Minimax-style policy optimization) methods have been proposed, though mainly in adversarial language alignment and not standard preference RLHF—thus falling outside our comparison.

## 2.3 COMPARISON TO OUR APPROACH

While prior work enhanced PPO variants via sampling tweaks, decoupled clipping, or value modeling, to our knowledge none have incorporated *multi-token action* in policy gradient optimization process as illustrated in Figure 1. Our **Multi-Token Policy Gradient Optimization (MPO)** modifies the sampling distribution in PPO and GRPO by aggregating importance sampling ratios over spans of $K$ future tokens. This block-level formulation yields a closer alignment between optimization and reasoning structures, and can provide more stable gradient updates in reasoning tasks. In contrast, traditional PPO and GRPO compute ratios per token; DAPO only adjusts clipping and sampling heuristics without changing the atomic unit of ratio computation. Similarly, VAPO remains token-based, emphasizing value modeling and credit assignment rather than redefining sampling granularity. Therefore, our method complements existing approaches by introducing a new block-level grouping of actions, which can be combined with other RLHF improvements.

## 3 PRELIMINARIES

We first review existing approaches to importance sampling in policy optimization—specifically PPO and its extensions such as GRPO—to clarify their formulation and highlight their limitations. This sets the stage for introducing our method, which extends importance sampling to account for longer-horizon behavior via multi-token predictions.

### 3.1 IMPORTANCE SAMPLING STRATEGY IN POLICY OPTIMIZATION

Proximal Policy Optimization (PPO) (Schulman et al., 2017) relies on importance sampling to support multiple epochs of minibatch updates with trajectories collected under a previous policy. The surrogate objective uses:

$$r_t = \frac{\pi_\theta(a_t \mid s_t)}{\pi_{\theta_{\text{old}}}(a_t \mid s_t)}, \tag{1}$$

and the clipped PPO loss is:

$$J(\theta) = \mathbb{E}_t \left[ \min \left( r_t \hat{A}_t, \ \text{clip}(r_t, \ 1 - \epsilon, \ 1 + \epsilon) \hat{A}_t \right) \right], \tag{2}$$

where $\hat{A}_t$ is the advantage estimate, and the clipping constrains large updates to maintain stability. Generalized Reward Policy Optimization (GRPO) (Shao et al., 2024) modifies the importance sampling ratio by introducing group-relative advantage sampling. At each state $s_t$, multiple candidate

actions $a_t$ are sampled and their empirical rewards $R_t$ are standardized:

$$\hat{A}_t = \frac{R_t - \bar{R}}{\sigma}, \tag{3}$$

where $\bar{R}$ and $\sigma$ are the mean and standard deviation across the group. The same importance ratio $r_t = \frac{\pi_\theta(a_t|s_t)}{\pi_{\theta_{\text{old}}}(a_t|s_t)}$ is used per sample, and the loss aggregates contributions averaged over sampled actions. Decoupled Clip and Dynamic Sampling Policy Optimization (DAPO) similarly builds on this framework by adapting the clipping threshold and employing rejection sampling, but does not alter the fundamental importance sampling ratio.

However, in all these methods, importance sampling is performed at the level of individual token steps—i.e., the ratio ($r_t$) compares local action probabilities at each timestep. In the context of language modeling, where accurate semantics and the maintenance of long-range dependencies depend on structural token blocks, such token-level ratios fail to capture behavioral differences or dependencies that span multiple steps. Besides, the variance of importance sampling estimators typically grows with increasing discrepancy between these policies or with longer trajectory lengths, since standard per-step ratio may become highly volatile (Metelli et al., 2020; Papini et al., 2024).

## 3.2 MULTI-TOKEN PREDICTION MECHANISM

To address the limited expressiveness of token-level representations, we turn to a multi-token representation mechanism that enables the model to jointly represent and predict short contiguous token sequences rather than treating each token independently. Current LLMs typically perform next-token prediction, modeling the probability

$$p(o_{t+1} \mid q, o_{1:t}) \tag{4}$$

with an autoregressive decoder, where q is the question prompt. Instead, the MTP implementation introduced in DeepSeek-V3/R1 (Liu et al., 2024) allows the model at each position $t$ to predict up to $K$ future tokens $o_{t+1}, o_{t+2}, \ldots, o_{t+K}$ through a sequence of MTP modules that preserve causal consistency (we provide detailed implementation of MTP modules in Appendix A). Specifically, the probability distribution on position $t + k$ is predicted as:

$$h_t^k = \text{TRM}_k(M_k[\text{Norm}(h_t^{k-1}); \text{Emb}(o_{t+k-1})]), \tag{5}$$

where $h_t^k \in \mathbb{R}^h$ is the hidden state of the $k^{th}$ MTP module, note that $h_t^0$ represent the hidden state output of the bacbone model for token $o_t$. $M_k$ is a learned projection layer, $\text{TRM}_k$ is a Transformer decoder block, and $\text{Emb}(o_{t+k})$ is the embedding of the future token $o_{t+k}$. Each module outputs with individual softmax prediction for the probability distribution of the $t + k + 1$ token of the sequence:

$$p(o_{t+k} \mid q, o_{1:t+k-1}) = \text{LM\_Head}(h_t^k), \tag{6}$$

for $k = 2, \ldots, K$. In supervised fine-tuning settings, the aggregated MTP objective is defined as:

$$\mathcal{L}_{\text{MTP}} = -\sum_{k=2}^{K} \alpha_k \log p(o_{t+k} \mid q, o_{1:t+k-1}). \tag{7}$$

The loss of the MTP module is assigned a decaying weight $\alpha_k$ to simulate the diminishing value of future information. Following previous works (Cai et al., 2024; Ankner et al., 2024), MPO first initialize the MTP modules using the last layer of the backbone model, then warm-up these modules using the objective described above to ensure the quality of multi-token prediction (see Appendix A and Figure 7(b) for implementation details). In this paper, we concatenate the train split of both GSM8K (Cobbe et al., 2021) and MATH (Hendrycks et al., 2021) as the warm-up data.

Although multi-token prediction has been adopted in various pre-training and fine-tuning settings to improve generation efficiency and performance, its integration into reinforcement-based post-training remains limited. Existing policy-gradient approaches, including PPO, GRPO, and DAPO, remain confined to token-level updates and overlook the structured, multi-step action patterns inherent to reasoning-oriented generation.

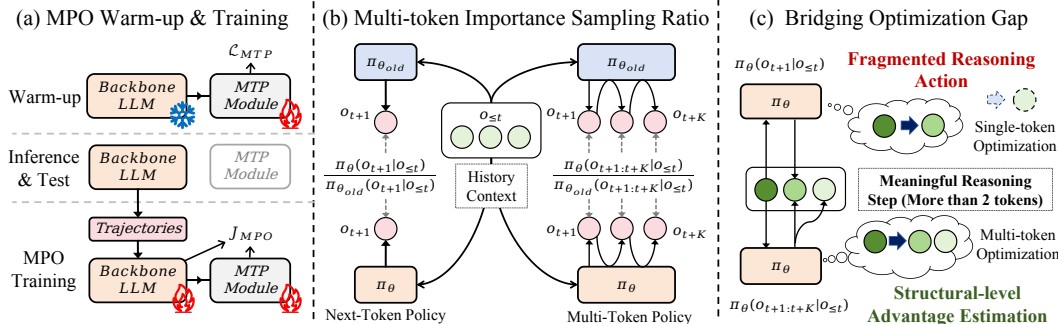

Figure 2: (a) Demonstration of the implementation of MPO warm-up and training process; (b) illustration of the united importance sampling ratio proposed in MPO method; (c) Comparison between single-token and multi-token optimization, multi-token optimization jointly models contiguous tokens as a structural reasoning action, enabling structural-level advantage estimation.

## 4 FUTURE KNOWLEDGE INJECTED OBJECTIVE

Building on the limitations of token-level optimization discussed in Section 3, we now present a new policy objective that explicitly incorporates future information into the update process.

### 4.1 FROM SINGLE-TOKEN TO MULTI-TOKEN

Let $q_i$ denote the prompt (input) for trajectory $i$, and $o_{i,1:t}$ denote the sequence of generated tokens up to position $t$ for trajectory $i$. The usual per-token importance sampling ratio $r_{i,t}$ for trajectory $i$ at position $t$ is defined as:

$$r_{i,t}(\theta) \;=\; \frac{\pi_\theta\big(o_{i,t+1} \mid q_i, o_{i,1:t}\big)}{\pi_{\theta_{\text{old}}}\big(o_{i,t+1} \mid q_i, o_{i,1:t}\big)}. \tag{8}$$

With the help of multi-token prediction modules, we are able to replace the importance sampling ratio with an aggregated ratio over spans of length K (as shown in Figure 2(b)):

$$r_{i,t}(\theta) \;=\; \frac{\pi_\theta\big(o_{i,t+1:t+K} \mid q_i, o_{i,1:t}\big)}{\pi_{\theta_{\text{old}}}\big(o_{i,t+1:t+K} \mid q_i, o_{i,1:t}\big)}. \tag{9}$$

The new objective then sums (or clips) these block-level ratios times advantages. This encourages coherent multi-token patterns rather than independent next-token moves. Essentially, through this way, we are able to leverage the multi-token prediction loss to enhance target LLM's future planning ability in the post-training process instead of the pre-training process which requires much higher computational costs. For example, when constructing the deepseek-r1 reasoning model , the pre-training stage costed around 90% of the total cost and the post-training (supervised fine-tuning and then reinforcement learning alignment) costed the remaining 10% budget. For those pretrained LLMs that did not use MTP loss during its pre-training process, we provide a solution to update the model's multi-token prediction ability in a post-training way.

### 4.2 MULTI-TOKEN POLICY-GRADIENT OBJECTIVE

Let each trajectory $i$ produce a sequence of generated tokens using original auto-regressive strategy:

$$o_i = \big(o_{i,1}, o_{i,2}, \ldots, o_{i,|o_i|}\big). \tag{10}$$

After generating the base sequence $o_{i,1:t}$ for prompt $q_i$, we compute the multi-token importance sampling ratio by sampling $K$ additional tokens in an auto-regressive fashion. Specifically, at each position $t$, for each $n \in [1, K]$, the generation of token $o_{i,t+n}$ is conditioned on the entire prefix $o_{i,1:t+n-1}$, meaning each new token is predicted based on all previously generated tokens, including those just sampled within the span. The revised multi-token importance sampling ratio is thus

formulated as:

$$R_{i,t}^{(K)}(\theta) = \prod_{n=1}^{K} \frac{\pi_\theta(o_{i,t+n} \mid o_{i,1:t+n-1})}{\pi_{\theta_{\text{old}}}(o_{i,t+n} \mid o_{i,1:t+n-1})}. \tag{11}$$

Though equation 11 can effectively reduce bias by taking more future actions into consideration (see mathematical derivations in Appendix B.2), such a production form of the importance ratio tends to dramatically increase the variance of sampling ratios, which further leads to a large clip fraction and hinders the optimization process in practice. Inspired by the Log-COP-TD method (Hallak & Mannor, 2017), we adopt an alternative trade-off formulation to control variance by replacing the original product of ratios with a weighted log-sum aggregation:

$$\widetilde{R}_{i,t}^{(K)}(\theta) = \exp\left(\sum_{n=1}^{K} \beta_n \log \frac{\pi_\theta(o_{i,t+n} \mid o_{i,1:t+n-1})}{\pi_{\theta_{\text{old}}}(o_{i,t+n} \mid o_{i,1:t+n-1})}\right), \tag{12}$$

where $\beta_n$ are non-negative step-wise weights satisfying $\sum_{n=1}^{K} \beta_n = 1$. Based on the weight of the first MTP module, denoted as $\beta_2$, we manually define

$$\beta_k = \beta_2 \times \lambda^{k-2}, \quad k \geq 2; 0 \leq \lambda \leq 1, \tag{13}$$

where $\lambda$ is a hyperparameter controlling the rate of future-information decay. This formulation applies a decaying weight to the $(k-1)^{\text{th}}$ MTP module, thereby incorporating the influence of future information with diminishing strength. This design prioritizes the impact of nearer-future predictions while still incorporating longer-horizon contributions in a controllable manner. We then define the policy-gradient surrogate objective, analogous to PPO but using the $K$-step ratio at each position $t$:

$$J_{MPO}(\theta) = \mathbb{E}_{q_i \sim D, \{o_i\}_{i=1}^G \sim \pi_{\theta_{old}}(.\mid q_i)}$$
$$\left[\frac{1}{\sum_{i=1}^{G} |o_i|} \sum_{i=1}^{G} \sum_{t=1}^{|o_i|} \min\left(\widetilde{R}_{i,t}^{(K)}(\theta)\, \hat{A}_{i,t},\ \text{clip}\big(\widetilde{R}_{i,t}^{(K)}(\theta), 1 - \epsilon_{low}, 1 + \epsilon_{high}\big)\hat{A}_{i,t}\right)\right], \tag{14}$$

where

$$\widetilde{R}_{i,t}^{(K)}(\theta) = \exp\left(\sum_{n=1}^{K} \beta_n \log \frac{\pi_\theta(o_{i,t+n} \mid o_{i,1:t+n-1})}{\pi_{\theta_{\text{old}}}(o_{i,t+n} \mid o_{i,1:t+n-1})}\right),$$

and $\hat{A}_{i,t}$ is the estimated advantage at position $t$. By jointly observing $K$ consecutive tokens in a single step, the estimator encourages optimization to account for a broader, structurally coherent view at the level of multi-token reasoning chunks (as illustrated in Figure 2(c)). In this paper, we focus on the experiments with $G = 1$ and we set $\epsilon_{low} = \epsilon_{high}$. We also conduct experiments with MTP-based value function in MPO, see Appendix C.

## 5 EXPERIMENTAL SETTINGS

In this section, we provide a brief overview of our experimental setup, including the baselines, evaluation methods, and training hyperparameter configurations.

### 5.1 DATASETS AND EVALUATION

**Model and Datasets.** We implement MPO on three widely used instruction-tuned backbone models: Llama3.2-1B-Instruct (Dubey et al., 2024), DeepSeek-Distilled-Qwen2.5-1.5B, and DeepSeek-Distilled-Qwen2.5-7B (Guo et al., 2025), to evaluate its effectiveness across different architectures and model scales. These backbones have been trained on diverse tasks that include mathematical reasoning. We assess MPO on two mathematical reasoning benchmarks of varying difficulty, GSM8K (Cobbe et al., 2021) (college-level) and MATH (Hendrycks et al., 2021) (competition-level), and further evaluate its coding capability on the HumanEval benchmark (Chen, 2021) to test cross-domain generalization. For HumanEval, we use the data provided by coding benchmark MBPP (Austin et al., 2021) as training set. More details can be found in Appendix A.2.

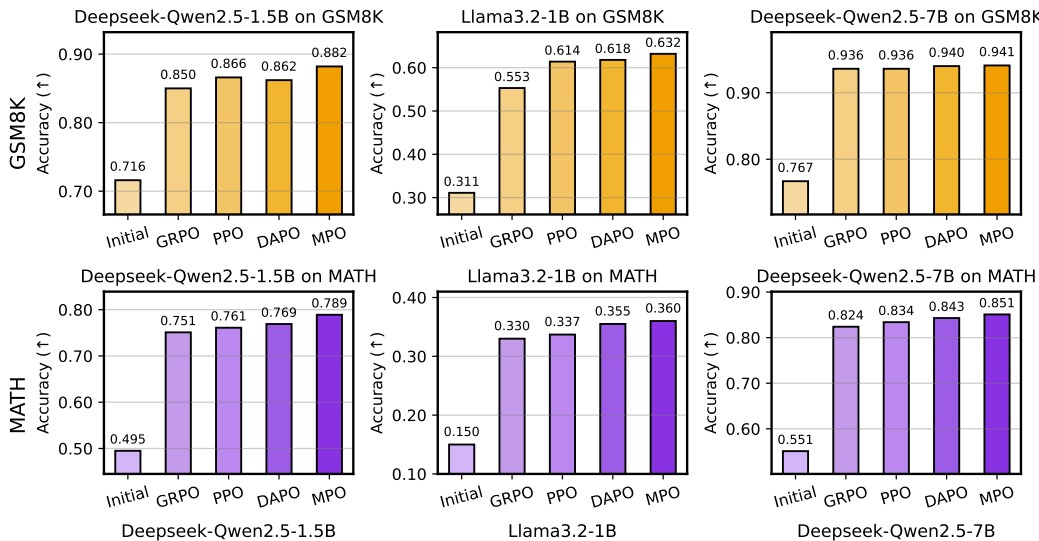

Figure 3: The performance of proposed MPO and baseline methods. MPO outperforms the baselines in most scenarios, demonstrating the effectiveness of aggregating block-wise information.

**Evaluation.** We adopt the pass@1 metric to evaluate the models' accuracy in answering mathematical questions and coding completions, which focuses on the correctness of the final answer provided by the model upon completion of the reasoning process. For math problems, to unify the evaluation process, we use the answering format of `"\boxed{answer}"`. As baselines, we select the most commonly used token-wise policy optimization strategies: PPO (Schulman et al., 2017), GRPO (Shao et al., 2024) and DAPO (Yu et al., 2025). In this paper, we mainly discuss the implementation and results of MPO based on PPO.

### 5.2 IMPLEMENTATION DETAIL AND HYPERPARAMETERS

As mentioned in section 3.2, we adopt a cold-start setting for both our proposed method and baselines, only warming up the MTP module before the training process of MPO. Regarding the hyperparameters introduced by the multi-token module, we report in Table 1 a representative sets of configurations in practice when $K = 2$. See Appendix A for more implementation details.

| Hyperparameter | Value |
|---|---|
| MTP warmup loss weight $\alpha$ | 0.4 |
| Weight of $1^{st}$ MTP module $\beta_2$ | 0.08 |
| MTP weight decay rate $\lambda$ | 0.9 |

Table 1: Typical Hyperparameter settings.

### 6 RESULTS

In this section, we analyze the task performance of MPO and effects of introducing future information and the training efficiency.

### 6.1 TASK PERFORMANCE

As shown in Figure 3, the proposed method consistently outperforms the baseline approaches—including PPO, GRPO and DAPO—across both GSM8K and MATH benchmarks under both evaluated model architectures and scales. This improvement demonstrates that optimizing over block-level semantic actions enables the model to maintain better structural coherence and long-horizon consistency during multi-step reasoning. On the HumanEval benchmark (as reported in Table 2), MPO also delivers consistent gains over PPO, GRPO, and DAPO, confirming that its advantages extend beyond symbolic reasoning to program synthesis tasks. The improvement indicates

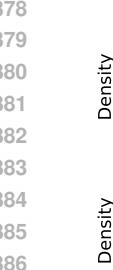
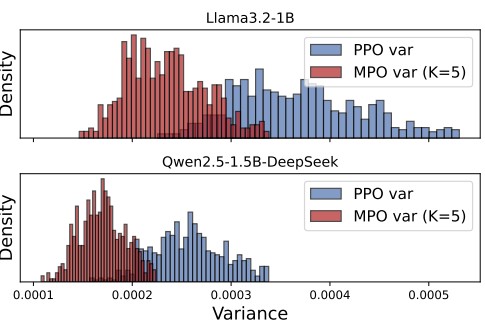
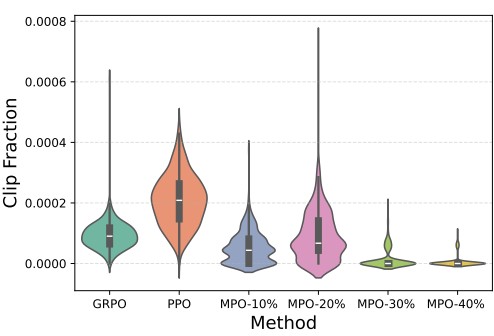

(a) Variance of importance sampling ratio.

(b) IS Ratio Clip fraction among strategies.

Figure 4: Comparison of variance in importance sampling ratios and clip fraction during training on the GSM8K dataset. Our MPO method significantly reduces the variance of the importance sampling ratios and lowers the clip fraction compared to PPO.

| Model | Zero-Shot | GRPO | DAPO | PPO | MPO (Ours) |
|---|---|---|---|---|---|
| Llama3.2-1B-Instruct | 0.354 | 0.372 | 0.396 | 0.390 | **0.403** |
| DeepSeek-Qwen2.5-1.5B | 0.451 | 0.591 | 0.603 | 0.598 | **0.640** |
| DeepSeek-Qwen2.5-7B | 0.689 | 0.811 | 0.817 | 0.805 | **0.841** |

Table 2: The performance of proposed MPO and baseline methods on coding task HumanEval, MPO consistently outperforms the baselines, which demonstrates the effectiveness of aggregating block-wise actions in advantage estimation process.

that modeling block-level actions benefits not only mathematical derivation but also structured code generation, where preserving multi-step semantic dependencies is similarly critical.

## 6.2 ANALYSING THE EFFECT OF INTRODUCING MULTI-TOKEN INFORMATION

**Q1: Does incorporating future information stabilize the training process?** As discussed in section 4.2, incorporating block-wise information broadens the optimization horizon of each action and can reduce bias from mathematical aspects. This is especially useful in reasoning tasks involving mathematical formulas or coding tasks with structural token blocks. Empirically, we observe that MPO stabilizes the importance sampling ratio. In Figure 4a and Figure 4b, we analyse the variance of importance sampling ratios on GSM8K. According to the results, by integrating future information, MPO consistently lowers the variance of impor-

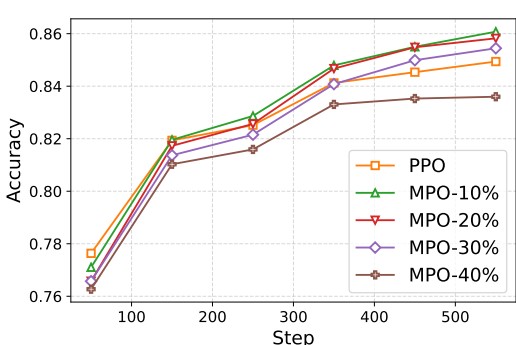

Figure 5: Varying future information weights.

tance sampling ratio and clip fraction during policy optimization, and this reduction becomes more pronounced as the weights $\beta_k$ from MTP modules increases. An open question, however, is how increasing the contribution of MTP weights influences learning dynamics and final performance.

**Q2: How much future information is appropriate to incorporate?** As shown in Figure 5, we investigate the effect of varying the proportion of future information incorporated through MTP modules on the GSM8K dataset using the DeepSeek Distill Qwen2.5-1.5B model. The overall contribution of future information is controlled by adjusting the cumulative weights of the MTP modules, i.e., the sum of $\beta_2$ to $\beta_K$, with a fixed decay factor $\lambda = 0.8$ in Eq. 13. The results reveal a trade-off between training stability and optimization bias. Increasing the proportion of future information initially stabilizes policy updates by reducing the variance of importance sampling ratios and the frequency of gradient clipping. However, when the proportion becomes too large, the ratio

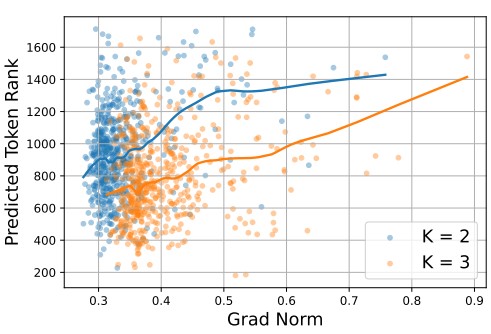

(a) Predicted token rank with avg. gradient norm.

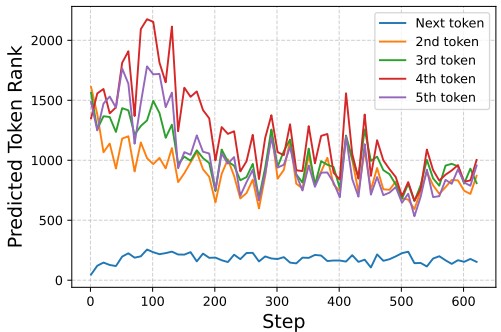

(b) Predicted token rank with training steps.

Figure 6: Reliability analysis of the incorporated future information from the MTP module. (a) Larger prediction errors correlate with higher gradient magnitudes. (b) Progressive improvement in the MTP module's imitation of the MTP module's next-K-token behavior.

| Dataset | MPO Task Accuracy | | |
|---|---|---|---|
| | K=2 | K=3 | K=5 |
| GSM8K | 0.871 | 0.875 | **0.882** |
| MATH | 0.771 | 0.753 | **0.789** |

(a) Analysis of the number of MTP modules.

| MTP Weight $\beta_2 =$ | 0.08 | 0.06 | 0.04 |
|---|---|---|---|
| $\lambda = 1.0$ | 0.745 | 0.773 | **0.787** |
| $\lambda = 0.9$ | 0.756 | 0.785 | 0.760 |
| $\lambda = 0.8$ | **0.758** | **0.789** | 0.764 |

(b) Grid Search of beta and decay rate.

Table 3: Effect of MTP block size $K$ and decay rate $\lambda$ on training stability and performance. Results from a grid search show that extending the decision horizon up to $K = 5$ and applying a moderate decay $\lambda = 0.8$ produces the most stable and effective configuration.

approximation method adopted and the noise from future token prediction may introduce additional bias, weakening the alignment between gradient updates and the next-token action objective. Empirically, the best performance is obtained when about 10–20% of the training signal originates from MTP modules, balancing stability and optimization effectiveness.

**Q3: Is the incorporated future information reliable throughout the training process?** Building on the previous analysis (Q2), where excessive MTP weights were found to reduce overall performance, we further examine whether such instability arises from unreliable or noisy future information. We use the logit rank of the predicted token—its position within the backbone model's vocabulary distribution—as a proxy for prediction accuracy. A smaller rank indicates closer alignment between the MTP module and the backbone's future-token distribution. Figure 6a(a) shows the relationship between the $2^{nd}$-token prediction rank and the average batch gradient norm under $K = 2$ and $K = 3$. Higher ranks correlate with larger gradient norms, suggesting that less accurate future predictions can indeed amplify gradient noise and destabilize training. Figure 6b(b) further illustrates that, although only warmed up initially, the MTP module's predictions improve steadily during training even without explicit entropy regularization. While a gap remains relative to the backbone's next-token accuracy, within a moderate range of integration, the impact of prediction noise introduced by future information in the current implementation of MPO remains controlled.

**Q4: How do the block size $K$ and decay rate $\lambda$ shape the bias–variance trade-off in MPO?** Having established the stability and reliability of MPO under moderate integration of future information, we further perform a grid search over the number of future tokens $K$ and the decay rate $\lambda$ to investigate their influence on training stability and optimization performance. Although expanding $K$ theoretically reduces bias by enabling longer-horizon decisions, it also increases variance and can destabilize training. As shown in Table 3a, performance improves as $K$ increases up to 5, indicating that extending the decision horizon is beneficial within this range, while larger $K$ values tend to amplify noise. To further examine the bias–variance trade-off, Table 3b reports the results of varying the decay rate $\lambda$ and the cumulative MTP weight $\sum_{k=2}^{K} \beta_k$. In all experiments, we set a relatively small initial weight $\beta_2$ so that the total contribution of future information remains below 30%, which stabilizes training while allowing meaningful future context. Incorporating a decay term for more distant predictions improves stability, and the configuration $\beta_k = \beta_2 \times 0.8^{k-2}$ consistently

outperforms using uniform weights ($\beta_k = \beta_2$). These results suggest that appropriately discounting farther predictions helps constrain variance while maintaining sufficient horizon extension, providing a practical balance between bias reduction and training stability in MPO.

### 6.3 DISCUSSION OF TRAINING EFFICIENCY

**Parameter Overhead.** Table 4 and Table 5 together summarize the resource overhead and comparative efficiency of MPO. During MPO training, the multi-token prediction (MTP) module is implemented following the efficient design adopted in DeepSeek-V3(Liu et al., 2024). The additional parameters introduced by MTP are used solely during training to estimate future-token policy ratios and are detached during inference, ensuring a fair comparison with baseline methods.

**Training and memory overhead.** Besides, the overhead of MPO is moderate in practice. When trained using fp16 precision on a single node with $8\times$ NVIDIA A100 (80 GB), the memory consumption of MPO ($K = 5$) is comparable to that of GRPO (31.0% vs. 30.6%), while training runs approximately 30% faster on average. Despite this computational overhead for an additional forward pass used to compute multi-token policy ratios after obtaining full trajectories, MPO delivers consistent accuracy improvements on both GSM8K and MATH benchmarks over PPO and GRPO. These results indicate that the added MTP modules contribute minimal training burden while offering substantial efficiency and performance benefits.

| Training Speed (Second per iteration) | | | |
|---|---|---|---|
| Model | PPO | K=3 | K=5 |
| Deepseek-Qwen2.5-1.5b | 25.6 | 30.3 | 33.3 |
| Llama3.2-1b | 19.1 | 25.0 | 28.3 |
| Model Size (Compared to the original model) | | | |
| Model | PPO | K=3 | K=5 |
| Deepseek-Qwen2.5-1.5b | 1.0$\times$ | 1.20$\times$ | 1.41$\times$ |
| Llama3.2-1b | 1.0$\times$ | 1.52$\times$ | 2.05$\times$ |

Table 4: The size of plug-and-play MTP modules and average training time per step.

| Method | Memory Usage | Training Time |
|---|---|---|
| PPO | 22.1% | 25.6 s |
| GRPO | 30.6% | 43.7 s |
| MPO | 31.0% | 33.1 s |

Table 5: Comparison of three methods in terms of average memory usage and training speed. Here we set group size=4 for GRPO and K=5 for MPO.

**Design of Warmup Stage.** To ensure stable optimization, we include a brief warm-up phase to pre-train the MTP modules (approximately 12 minutes for one epoch) before the MPO fine-tuning stage (around 7 hours for 20 epochs), incurring only a lightweight additional time cost. This design choice follows prior multi-head decoding frameworks(Cai et al., 2024; Ankner et al., 2024) and represents a standard stabilization step rather than additional complexity unique to our method.

Overall, MPO emphasizes a modular and practical architecture, this design enables the framework to improve optimization stability and reasoning performance; furthermore, future work may explore even lighter-weight MTP variants for large-scale deployment (also discussed in Appendix D).

## 7 CONCLUSION

In this work, we proposed Multi-Token Policy Gradient Optimization (MPO), a general framework that extends traditional token-level policy optimization to block-level semantic actions through multi-token prediction (MTP) modules. This design bridges the gap between local token decisions and the structured, long-horizon reasoning behavior required by complex language tasks. Extensive experiments on GSM8K, MATH, and HumanEval show that MPO consistently outperforms standard policy-gradient baselines such as PPO, GRPO, and DAPO. Further analyses demonstrate that moderate incorporation of future information stabilizes training by reducing gradient variance, that there exists an optimal integration ratio balancing bias and variance, and that multi-token prediction noise remains controllable within this range. Finally, systematic exploration of the block size and decay rate reveals their complementary roles in optimizing this trade-off. MPO offers a modular, scalable extension to existing RLHF frameworks, introducing moderate overhead while improving optimization stability and reasoning performance. We believe this work provides new insights into the design of fine-grained importance-sampling strategies and opens a promising direction for future research on structured and long-horizon optimization in large language models.

## REPRODUCIBILITY STATEMENT

We will briefly go through the tools and data we use in this paper for reproducibility.

**MTP Modules.** We developed our Multi-token Prediction modules based on transformers (ver 4.51.1) and trl (ver 0.22.0), both frameworks are open-sourced by the huggingface community. We modify the modeling library file of the transformers framework to achieve multi-token prediction.

**MPO Approach.** The overall MPO approach is developed based on the open-source reinforcement learning framework Verl (volcano engine, ver 0.3.1). We modify the PPO Trainer Class and Actor workers Class library to compute and gather multi-token log probability distributions when computing the policy loss of MPO.

**Datasets and models.** We conduct our experiments with open-source datasets GSM8K and MATH for MTP warmup and MPO training. The two models are also from the open-source community.

**Code and implementations.** In the supplementary materials, we include a detailed README file that guides the reproduce process, and we provide our implementation of the causal MTP module as well as the core code computing MPO policy loss for reference.

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

## A  IMPLEMENTATION DETAILS

We will provide separate explanations of the overall MPO implementation process, the experimental parameter settings, and details of the code and hardware implementation.

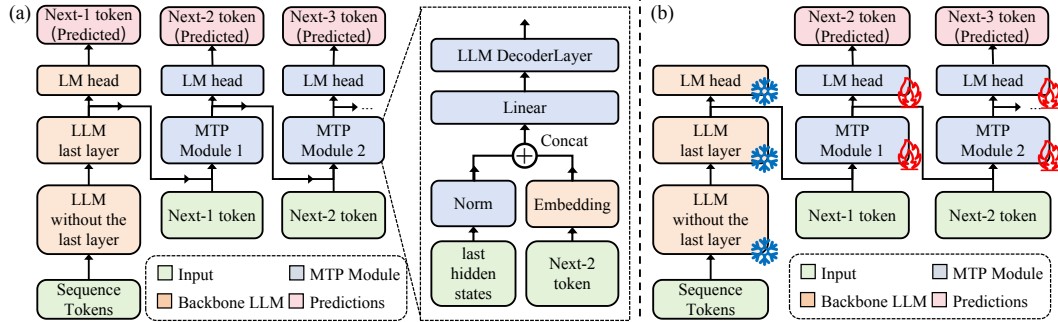

Figure 7: (a) The implementation of MTP module used in MPO, the structure preserve causal consistency following Deepseek-V3; (b) Demonstration of MPO warmup stage to initialize MTP Modules before RL stage.

### A.1 MTP MODULES

The implementation of MPO requires the MTP module to predict the model's policy distributions at multiple positions along a given trajectory $i$, enabling the calculation of importance sampling coefficients over blockwise sequences. Achieving stable multi-position decision making depends on the MTP module's ability to produce accurate predictions of future policy distributions. These distributions cannot be directly modeled using only the backbone's final layer and language output head, so an appropriate warmup is necessary. Although previous works has noted that MTP can improve the model's reasoning ability during fine-tuning (Liu et al., 2024), to ensure fair comparisons, we do not backpropagate gradients to the backbone during MPO finetuning and only train the MTP module (as shown in Figure 7), which has also been adopted in previous efficient MTP decoding works (Cai et al., 2024). As we discussed previously, each MTP module and its corresponding LM head are deeply copied from the backbone model's final layer and LM head, respectively, in order to avoid an overly complex initialization process.

### A.2 DATASET AND TRAINING PARAMETERS

We conduct experiments using three widely used mathematical reasoning and coding benchmarks: GSM8K (Cobbe et al., 2021), MATH (Hendrycks et al., 2021) and HumanEval (Chen, 2021). For HumanEval, we combined the sanitized and original data provided in coding benchmark MBPP (Austin et al., 2021) to conduct MPO training. Table 6 summarizes the main statistics of these datasets. The question-answer pairs in both datasets are provided in a natural language format, suitable for language model-based reasoning experiments.

| Dataset | # Train Samples | # Test Samples | Avg. Sample Length |
|---|---|---|---|
| GSM8K | 7,473 | 1,319 | ∼80 tokens |
| MATH | 7,500 | 5,000 | ∼200 tokens |
| HumanEval | 1399 (MBPP) | 164 | ∼130 tokens |

Table 6: Statistics of datasets used for training and evaluation.

We shuffle and concatenate the training splits of two math datasets for MTP warm-up training. For MPO training, we conduct separate experiments on each dataset, using the corresponding training split for learning and the corresponding test split for evaluation. This setup allows us to evaluate the generalization and effectiveness of our method on both mathematical reasoning and coding tasks.

| Parameter | Value |
|---|---|
| Train batch size | 256 |
| PPO mini batch size | 128 |
| PPO micro batch size / GPU | 4 |
| Number of GPUs | 8 |
| Number of Nodes | 1 |
| Max prompt length | 1024 |
| Max response length | 2048 |
| Learning rate (actor) | $1 \times 10^{-6}$ |
| Learning rate (critic) | $1 \times 10^{-5}$ |
| Total epochs | 20 |

Table 7: Key hyperparameters for MPO training.

## A.3 MPO TRAINING PROCESS

---
**Algorithm 1:** Multi-Token Policy Optimization (MPO)

---
**Input** : Dataset $\mathcal{D}$ of prompts $q_i$ and annotated outputs $o'_i$
Pretrained LLM with MTP module; initial parameters $\theta$
Old policy parameters $\theta_{\text{old}}$
Block size $K$, weights $\alpha_k, \beta_k$; clip parameter $\epsilon_{low}, \epsilon_{high}$
**Output:** Updated policy parameters $\theta$

**Stage 1: MTP Warm-Up**
**for** *several epochs* **do**
 **for** *batch of* $(q_i, o'_i)$ *in* $\mathcal{D}$ **do**
  **for** *each position* $t$ *in* $o'_i$ **do**
   **for** $k = 1$ *to* $K$ **do**
    Predict $p(o'_{t+k} \mid q, o'_{1:t})$ using backbone model ($k = 1$) and MTP module
    ($k \geq 2$)
   Compute $\mathcal{L}_{\text{MTP}} = -\sum_{k=2}^{K} \alpha_k \log p(o'_{t+k} \mid q, o'_{1:t})$
  Update only MTP parameters by minimizing $\mathcal{L}_{\text{MTP}}$

**Stage 2: MPO Fine-tuning**
**for** *each MPO epoch* **do**
 $\mathcal{B} \leftarrow$ sample mini-batch of prompts $q_i$ from $\mathcal{D}$
 **for** *each* $q_i$ *in* $\mathcal{B}$ **do**
  Generate trajectory $o_i$ using $\pi_{\theta_{\text{old}}}$
  Compute per-token or group-based advantages $\hat{A}_{i,t}$
 **for** *each trajectory* $(q_i, o_i)$ *and each position* $t$ *in* $o_i$ **do**
  `// Sample K-step target block`
  Sample $o_{i,t+1:t+K}$ in an autoregressive manner using $\pi_{\theta_{\text{old}}}$
  Initialize $D_{kl} = 0$
  **for** $n = 1$ *to* $K$ **do**
   Compute:
   $p_\theta = \pi_\theta(o_{i,t+n}|o_{i,1:t+n-1})$
   $p_{\theta_{\text{old}}} = \pi_{\theta_{\text{old}}}(o_{i,t+n}|o_{i,1:t+n-1})$
   $D_{kl}\mathrel{+}= \beta_n \cdot (\log p_\theta - \log p_{\theta_{\text{old}}})$
  $\widetilde{R}_{i,t}^{(K)} = \exp(D_{kl})$
  Compute surrogate loss for $(i,t)$:
  $J_{i,t} = \min\left( \widetilde{R}_{i,t}^{(K)} \hat{A}_{i,t}, \ \text{clip}(\widetilde{R}_{i,t}^{(K)}, 1 - \epsilon_{low}, 1 + \epsilon_{high})\hat{A}_{i,t} \right)$
 Aggregate loss:
 $J_{\text{MPO}} = \frac{1}{N} \sum_i \sum_t J_{i,t}$
 Update parameters $\theta$ via gradient descent using $J_{\text{MPO}}$
 Optionally, update $\theta_{\text{old}}$ periodically

---

Here we provide additional implementation details for the Multi-Token Policy Optimization (MPO) method described above, including t he hyper-parameters and a pseudocode for clarity. Table 7 lists our main hyperparameters and settings for the MPO training process. These values are extracted from our training scripts and reflect typical settings for large-scale RLHF or policy optimization tasks on math reasoning datasets. For HumanEval, due to the extensive thinking behavior of Deepseek distilled models, we extend the response length to 3072.

# B DISCUSSION ON INCORPORATING MULTI-TOKEN INFORMATION

This appendix provides a formal discussion of why incorporating multi-token prediction into policy optimization can improve optimization performance and enhance long-horizon reasoning ability. The exposition is consistent with the notation and definitions in the main text (Sections 3.2–4).

## B.1 INTUITIVE EXAMPLE AND MOTIVATION

In standard policy optimization methods such as PPO, the policy improvement step is based on per-token decisions, where the importance ratio

$$r_{i,t}(\theta) = \frac{\pi_\theta(o_{i,t+1} \mid q_i, o_{i,1:t})}{\pi_{\theta_{\text{old}}}(o_{i,t+1} \mid q_i, o_{i,1:t})} \tag{15}$$

compares the likelihood of a single action (next token) under the updated and old policies. This formulation implicitly assumes that: 1) the reward structure can be well captured by local token-level updates, and 2) long-horizon dependencies are either negligible or already encoded in the advantage estimator $\hat{A}_{i,t}$.

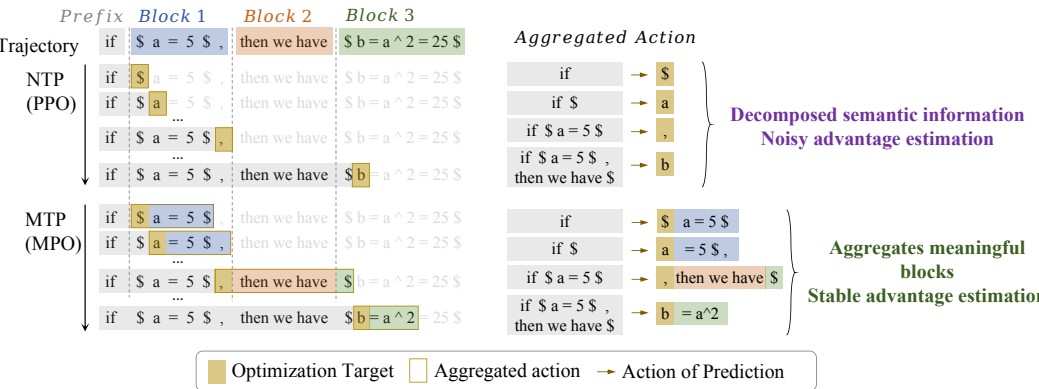

Figure 8: Illustration of token-level (NTP/PPO) vs. block-level (MTP/MPO) optimization. MTP aggregates tokens into semantically meaningful blocks for prediction and optimization, thereby better capturing sequence structure and long-range dependencies.

While traditional one-step token-level policy optimization (such as NTP/PPO, as shown in the upper section of Figure 8) updates the model based on individual token predictions, our illustration highlights that semantics and reasoning patterns in language generation typically emerge from coherent blocks of multiple tokens. As shown in the figure, generating a valid reasoning step or factual entity (e.g., $a=5$ or $b = a^2 = 25$) requires accurately predicting a sequence of tokens as a whole.

Limiting optimization to one-step advantage estimation introduces several issues:

1. **Fragmented learning signals**: Token-level updates isolate local token actions, weakening the internal consistency of reasoning segments.

2. **Incomplete credit assignment**: Single-token objectives struggle to capture dependencies whose effects span across multiple tokens within a reasoning step.

3. **Granularity mismatch**: The optimization signal operates at the token level, while the semantic correctness of reasoning typically emerges at the segment level.

In contrast, as illustrated in the lower MTP/MPO section, aggregating actions and optimization targets at the block level enables updates that better capture semantic integrity, reduce estimation variance, and align training with the goals of structured reasoning.

By instead defining a $K$-step decision ratio,

$$R_{i,t}^{(K)}(\theta) = \prod_{n=1}^{K} \frac{\pi_\theta(o_{i,t+n} \mid o_{i,1:t+n-1})}{\pi_{\theta_{\text{old}}}(o_{i,t+n} \mid o_{i,1:t+n-1})}, \tag{16}$$

we directly compare the joint likelihood of a future span under the new and old policies. This transforms the optimization from purely myopic next-token correction into a multi-step decision process, allowing the policy gradient to align with behaviors that unfold across multiple tokens. In essence, the surrogate objective is no longer tied to local moves but reflects higher-level planning, where corrections propagate over spans of length $K$ rather than being confined to single steps.

### B.2 THEORETICAL DERIVATION AND BIAS REDUCTION

#### B.2.1 PROBLEM SETUP AND NOTATION

We consider a standard reinforcement learning setup for sequence generation. Let time steps be indexed by $t = 0, 1, 2, \ldots$, where each step corresponds to generating a token.

- State: $s_t$ denotes the state at time $t$, which includes the context (previous tokens and hidden states).
- Action: $a_t$ denotes the token generated at time $t$.
- Policy: $\pi_\theta(a_t \mid s_t)$, with $\pi_{\text{old}}$ denoting the previous policy.
- Discount factor: $\gamma \in (0, 1]$.
- Value function: $V^\pi(s) = \mathbb{E}_\pi \left[ \sum_{k=0}^\infty \gamma^k r_{t+k} \mid s_t = s \right]$
- Advantage function: $A^\pi(s_t, a_t) = Q^\pi(s_t, a_t) - V^\pi(s_t)$
- Estimated value function: $\hat{V}_\phi(s)$

In standard one-step PPO, the advantage is often estimated using the TD error and generalized advantage estimator (GAE):

$$\delta_t = r_t + \gamma \hat{V}_\phi(s_{t+1}) - \hat{V}_\phi(s_t), \tag{17}$$

$$\hat{A}_t^{\text{GAE}} = \sum_{l=0}^\infty (\gamma\lambda)^l \delta_{t+l}, \tag{18}$$

where $\lambda \in [0, 1]$ controls the bias-variance trade-off.

#### B.2.2 K-STEP / MULTI-TOKEN CORE IDEA

In K-step (multi-token) prediction, at time $t$ we consider predicting not just $a_t$, but the next $K$ tokens jointly. Define the K-step return:

$$G_t^{(K)} := \sum_{k=0}^{K-1} \gamma^k r_{t+k} + \gamma^K \hat{V}_\phi(s_{t+K}). \tag{19}$$

The corresponding K-step advantage estimator is

$$\hat{A}_t^{(K)} := G_t^{(K)} - \hat{V}_\phi(s_t). \tag{20}$$

Intuitively, as $K$ increases, the advantage relies more on the actual observed rewards and less on the bootstrapped value function $\hat{V}_\phi$, thereby reducing the bias from value function approximation.

### B.2.3 BIAS ANALYSIS

Let $\epsilon_V := \sup_s \left| \hat{V}_\phi(s) - V^\pi(s) \right|$ denote the maximum value function approximation error. The K-step return based on the true value function is

$$G_t^{(K),\text{true}} := \sum_{k=0}^{K-1} \gamma^k r_{t+k} + \gamma^K V^\pi(s_{t+K}), \tag{21}$$

so the error due to $\hat{V}_\phi$ is

$$\Delta G_t^{(K)} := G_t^{(K)} - G_t^{(K),\text{true}} = \gamma^K(\hat{V}_\phi(s_{t+K}) - V^\pi(s_{t+K})). \tag{22}$$

Hence, in absolute value,

$$|\Delta G_t^{(K)}| \le \gamma^K \epsilon_V. \tag{23}$$

The bias of the K-step advantage relative to the true K-step advantage is

$$\hat{A}_t^{(K)} - A_t^{\text{true},(K)} = \left(G_t^{(K)} - \hat{V}_\phi(s_t)\right) - \left(G_t^{(K),\text{true}} - V^\pi(s_t)\right) \tag{24}$$

$$= \gamma^K(\hat{V}_\phi(s_{t+K}) - V^\pi(s_{t+K})) - (\hat{V}_\phi(s_t) - V^\pi(s_t)), \tag{25}$$

with absolute value bounded by

$$\left| \hat{A}_t^{(K)} - A_t^{\text{true},(K)} \right| \le \gamma^K \epsilon_V + \epsilon_V = (1 + \gamma^K)\epsilon_V. \tag{26}$$

If we focus on the relative dependence on one-step (n=1), note that the error term for one-step is $(1 + \gamma)\epsilon_V$. More importantly, if we decompose the bias into "the bootstrapping bias from the tail" and "the bias from the current estimate," we can see that the bootstrapping term from the tail is multiplied by $\gamma^K$.

In particular, when we consider the bias propagation with respect to the true infinite-horizon return, the bias term related to n-step bootstrapping decays as $\mathcal{O}(\gamma^K)$. In other words, the bias is introduced by bootstrapping (relying on $\hat{V}_\phi(s_{t+n})$), and its coefficient is $\gamma^K$. As $n$ increases, this term decays exponentially by a factor of $\gamma^K$, thus reducing reliance on $\hat{V}_\phi \Rightarrow$ reducing the bias introduced by function approximation.

### B.2.4 EXPECTED ADVANTAGE BIAS

Consider the expected bias over trajectories:

$$\mathbb{E}\left[\hat{A}_t^{(K)} - A^\pi(s_t, a_t)\right] = \mathbb{E}\left[G_t^{(K)} - \hat{V}_\phi(s_t) - (Q^\pi(s_t, a_t) - V^\pi(s_t))\right]$$

$$= \mathbb{E}\left[\sum_{k=0}^{K-1} \gamma^k r_{t+k} + \gamma^K \hat{V}_\phi(s_{t+K}) - \hat{V}_\phi(s_t) - \sum_{k=0}^{\infty} \gamma^k r_{t+k} + V^\pi(s_t)\right]$$

$$= \mathbb{E}\left[\gamma^K(\hat{V}_\phi(s_{t+K}) - V^\pi(s_{t+K})) - (\hat{V}_\phi(s_t) - V^\pi(s_t))\right.$$

$$\left. - \sum_{k=K}^{\infty} \gamma^k \left(r_{t+k} - \mathbb{E}[r_{t+k} \mid s_{t+K}]\right)\right]. \tag{27}$$

Under standard assumptions, the last term has zero expectation, so the dominant contribution to bias comes from

$$\mathbb{E}\left[\hat{A}_t^{(K)} - A^\pi(s_t, a_t)\right] \approx \mathbb{E}\left[\gamma^K(\hat{V}_\phi(s_{t+K}) - V^\pi(s_{t+K})) - (\hat{V}_\phi(s_t) - V^\pi(s_t))\right]. \tag{28}$$

If the bias of $\hat{V}_\phi$ at different time steps is not systematically correlated (or its expectation is approximately zero), then the main remaining error comes from the tail error scaled by $\gamma^n$, with the overall expected bias reduced to $\mathcal{O}(\gamma^n \epsilon_V)$. The expected bias due to reliance on the bootstrapped value in n-step is reduced to the order of $\gamma^n$, thereby lowering the bias.

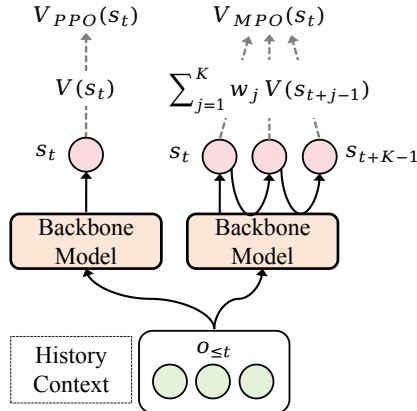

Figure 9: The implementation of value function on original auto-regressive backbone model (left) and those with MTP module (right). For multi-token prediction models, we try a weighted sum of the output from the shared value head.

## C  FUTURE VALUE ESTIMATION

### C.1  EXPERIMENTS WITH MULTI-TOKEN VALUE HEADS

In parallel with our multi-token perspective modification for the actor, we have also conducted experiments on the critic model, allowing it to anticipate the returns of actions over a future interval during training. To accommodate multiple multi-token prediction heads, we aggregate their outputs at each decision point via a learned convex combination:

$$V_{\text{MPO}}(s_t) = \sum_{j=1}^{K} w_j V_j(\hat{s}_{t+j-1}) \tag{29}$$

where $\hat{s}_{t+j-1}$ denotes the predicted future states and

$$w_j = \frac{\exp(\gamma_j)}{\sum_{k=1}^{K} \exp(\gamma_k)}, \quad \boldsymbol{\gamma} = (\gamma_1, \ldots, \gamma_K) \tag{30}$$

with $\gamma$ as learnable parameters, balancing the value botained from predicted future states. $V_j$ is the value head append to the $j^{th}$ MTP module. Given this value estimation, the advantage at each position $t$ is computed as

$$\hat{A}_{i,t} = \hat{Q}_{i,t} - V_{\text{MPO}}(s_t), \tag{31}$$

where $\hat{Q}_{i,t}$ is the estimated return as in standard policy gradient methods. This approach permits each value head to specialize, while the aggregation provides a flexible baseline for stable advantage calculation and policy optimization over multi-token predictions. Figure 9 illustrates the value estimation process.

### C.2  EFFECTS OF BLOCK SIZE $K$ AND FUTURE VALUE $V_{MPO}$

| Settings | GSM8K | | | MATH | | |
|---|---|---|---|---|---|---|
| | K=2 | K=3 | K=5 | K=2 | K=3 | K=5 |
| MPO | 0.871 | 0.875 | **0.882** | 0.771 | 0.753 | **0.789** |
| + $V_{MPO}$ | 0.879 | 0.875 | **0.882** | 0.773 | 0.756 | 0.762 |

Table 8: MPO performance with different token block size K and ablation study of future value estimation strategy.

We conducted experiments with Deepseek-Distill-Qwen2.5-1.5b by fixing the weight sum of MTP modules ($\sum_{k=2}^{K} \beta_k$) of future knowledge at 20%, with results shown in Table 8. The method

of injecting future information from the critic side did not improve overall performance as expected. Although our initial experiments suggested that this approach could have a "symmetric" effect—namely, by training the MTP module on the value prediction task to enhance overall training stability—this was not supported by later results. After addressing certain implementation issues in our code, it appears that MPO is not sensitive to the incorporation of future information into the value function. In fact, with K = 4 or 5, it may introduce greater instability and cause the model to oscillate at a suboptimal level. Simply injecting value predictions by summing the outputs of multiple MTP modules does not seem to effectively improve value estimation accuracy.

## D  FUTURE WORKS

Since MPO is developed based on MTP, and MTP is currently mostly used in pre-training and speculative decoding strategies, there is limited relevant research on its application in model post-training optimization. We find that this research topic still offers many avenues worth exploring. Below, we briefly discuss some directions for future work:

**Warmup Strategies.** In the DeepSeek V3 paper and subsequent work, we observe that MTP-based training can improve model reasoning performance. However, during the post-training phase—especially when training models that have not undergone MTP-based pre-training (such as Llama3 and Qwen2.5)—joint training can sometimes even reduce the final performance achieved through subsequent reinforcement learning optimization. In MPO, we address this conflict by freezing the backbone model during training, but this remains an open research question deserving further investigation.

**High-Performance MTP Implementation.** MTP technology is computationally intensive, and memory usage increases significantly with the number of MTP modules. In MPO, we preliminarily used a setting with up to K=5. However, the original MTP paper suggests that using even more MTP modules (such as 8) may yield better performance. How to efficiently implement these future information prediction strategies is a promising direction for further research.

**Efficient MTP on Scaled Models.** As discussed in the main text, MTP increases the overall model size, and the additional parameters scale up along with the backbone model. Although this overhead arises from the MTP technique itself, it hinders efficient experimentation with MPO on models of 7B parameters and beyond. Developing more efficient MTP mechanisms would greatly improve the practicality of MPO and enable us to further investigate its performance impact in large-scale settings.

## E  USAGE OF LARGE LANGUAGE MODELS

During the writing of this paper, large language models were used solely as tools for:

- **Grammar checking**: Asking the LLM to check whether there is any grammar mistakes in the given paragraph.
- **Translation**: Asking the LLM to provide a proper translation for certain phrases and expressions.
- **Readability improvement**: Asking the LLM to give suggestions of how to change the tone of our written paragraphs for fluency.

