# OpenReview forum: "Multi-Token Policy Gradient Optimization"
_ICLR.cc/2026/Conference — ICLR 2026 Conference Withdrawn Submission_

### Official Review · Reviewer_fG3C · 2025-10-15

**Soundness:** 2
**Presentation:** 3
**Contribution:** 2
**Rating:** 4
**Confidence:** 4

**Summary:**

This paper focuses on the challenge of granularity mismatch between sequence-level rewards and token-level actions in auto-regressive language models. The authors propose Multi-token Policy Gradient Optimization (MPO), a method that addresses this by aggregating contiguous blocks of future tokens into unified action units, rather than treating each token as an isolated action. The method is evaluated on the GSM8K and MATH benchmarks.

**Strengths:**

- This paper is generally well-written and easy to follow.
- The idea of treating semantically continuous tokens as a group for optimization is novel and intuitively sound: disadvantageous token spans should be suppressed collectively, while advantageous ones should be encouraged as a whole.
- This paper tries to address the important challenge of the granularity mismatch between token-level optimization and sequence-level reward signals in current RLVR scenario.

**Weaknesses:**

- **The motivation behind this paper is somewhat confusing**

While MPO is proposed to address the granularity mismatch between token-level generation and sequence-level rewards by aggregating tokens into blocks, a mismatch still remains between these block-level units and the sequence-level reward. The authors should clarify how this approach truly resolves the core issue.

- **Lack of Justification for hyperparameter K**

The paper's motivation for MPO is based on grouping semantically coherent segments, as illustrated in Figure 1(a) in the paper, where segments like equations vary in length. For MPO to be most effective, one would expect it to identify the natural boundaries of these semantic units. However, as described in Section 4, the method does not appear to identify such boundaries and instead relies on a fixed hyperparameter K to evenly divide the token sequence into contiguous blocks.

A related question concerns the optimal value of K. Table 2(a) indicates that K=5 yields the best performance among the values {2,3,5}. This seems to contradict the intuitive example in Figure 1(a), where the authors state that the "model's decision-making process needs to switch constantly between long token segments with more than 10 tokens." Could the authors explain this apparent mismatch? Specifically, why does a relatively small K=5 achieve optimal performance when the semantic segments it is meant to approximate are typically much longer?

- **Limited experimental evaluation**

The experimental evaluation is limited to two mathematical reasoning benchmarks, GSM8K and MATH. This raises concerns about the generalizability of MPO's effectiveness. To robustly support the claim that MPO is effective for "large language model post-training," additional evaluations on diverse tasks (e.g., code generation and instruction following) are necessary.

**Questions:**

In section 6.2, the authors state, "(clip fraction) reduction becomes more pronounced as the proportion of future information increases. This suggests that future-aware training stabilizes policy updates" (lines 405-407). However, on lines 417-419, they state, "This highlights that excessive incorporation of future information may lead to training instability." Both experiments were conducted using the same settings (MPO-10%/20%/30%/40%) but appear to have contradictory conclusions. I hope the authors can clarify this for me.

---

> ### Author Response · Authors · 2025-11-26
> **Response to Reviewer fG3C Part 1/2**
>
> > **Weakness 1:** The motivation behind this paper is somewhat confusing. While MPO is proposed to address the granularity mismatch between token-level generation and sequence-level rewards by aggregating tokens into blocks, a mismatch still remains between these block-level units and the sequence-level reward. The authors should clarify how this approach truly resolves the core issue.
>
> ---
>
> **Response to Weakness 1:**
>
> Your understanding is correct: the aggregation in MPO is performed at the block level rather than the sequence level. MPO aims to address the granularity mismatch between token-level optimization and sequence-level rewards by leveraging multi-token joint ratios to capture multi-step decision information, thereby theoretically achieving the aggregation of more complete semantic decision dependency information.
>
> In this paper, our use of "sequence level" was intended as an extension relative to the token level, which may have caused misunderstandings. There is indeed still a granularity gap between block and sequence, but we believe that directly optimizing sequence-level rewards remains challenging (e.g., due to the noise introduced during long-range decision aggregation).
>
> Thank you for your comments. We will revise the expression of "sequence level" in the updated version to avoid misunderstandings.
>
> ---
>
> > **Weakness 2:**
> The paper's motivation for MPO is based on grouping semantically coherent segments, as illustrated in Figure 1(a) in the paper, where segments like equations vary in length. For MPO to be most effective, one would expect it to identify the natural boundaries of these semantic units. However, as described in Section 4, the method does not appear to identify such boundaries and instead relies on a fixed hyperparameter K to evenly divide the token sequence into contiguous blocks.
>
> > A related question concerns the optimal value of K. Table 2(a) indicates that K=5 yields the best performance among the values {2,3,5}. This seems to contradict the intuitive example in Figure 1(a), where the authors state that the "model's decision-making process needs to switch constantly between long token segments with more than 10 tokens." Could the authors explain this apparent mismatch? Specifically, why does a relatively small K=5 achieve optimal performance when the semantic segments it is meant to approximate are typically much longer?
>
> ---
>
> **Response to Weakness 2:**
>
> This is a misunderstanding. During training, MPO uses the joint ratio over the next K tokens **at every position** to update the advantage estimate, rather than statically dividing the entire sequence into segments of length K. The value of K determines how many future tokens the model is allowed to incorporate at each position.
>
> According to our experimental results (see Table 4 in Appendix B.2), increasing K can effectively improve optimization performance. We will further explore more efficient engineering implementations of MTP to enable the integration of decision information over even longer sequences.
>
> Thank you for your comment. We hope that these clarifications can clear up the misunderstanding.

---

> ### Author Response · Authors · 2025-11-26
> **Response to Reviewer fG3C Part 2/2**
>
> > **Weakness 3:** Limited experimental evaluation
> The experimental evaluation is limited to two mathematical reasoning benchmarks, GSM8K and MATH. This raises concerns about the generalizability of MPO's effectiveness. To robustly support the claim that MPO is effective for "large language model post-training," additional evaluations on diverse tasks (e.g., code generation and instruction following) are necessary.
>
> ---
>
> **Response to Weakness 3:**
>
> Following previous works [1,2], we conducted evaluations on two widely used mathematical datasets of different difficulty levels, GSM8K and MATH, comparing our proposed MPO method with the baseline PPO algorithm and the widely used GRPO algorithm. These empirical results can, to some extent, demonstrate the effectiveness of our approach.
>
> In addition, following your suggestion, we conducted experiments on the HumanEval code dataset to the best of our ability. Based on the MTP model warmed up on mathematical reasoning tasks, we directly used the MBPP dataset as the training set (with a total of 1,399 training samples) and carried out 20 epochs of MPO training. We also reproduced the baseline methods under the same experimental settings; the results are shown in the table below:
>
> | Model| zero-shot | GRPO |PPO | MPO |
> |--------|--------------------------|------------------------------------|-------|-------|
> | Deepseek-qwen-1.5b | 0.451 |0.591| 0.598 | **0.640**(↑4.2%) |
> | llama3.2-1b | 0.354 |0.372| 0.390 | **0.403** (↑1.3%)|
>
> We have indicated the performance improvement of MPO over PPO. Based on these results, we observe that our proposed MPO achieves significant performance gains over PPO and GRPO in code tasks on both models. This further demonstrates the effectiveness of our method.
>
> Thank you for your suggestion; we will update the paper with these results.
>
> ---
>
> [1] Kazemnejad, Amirhossein, Milad Aghajohari, Eva Portelance, Alessandro Sordoni, Siva Reddy, Aaron Courville, and Nicolas Le Roux. "VinePPO: Refining Credit Assignment in RL Training of LLMs." In Forty-second International Conference on Machine Learning. 2024.
>
> [2] Wang, Peiyi, Lei Li, Zhihong Shao, Runxin Xu, Damai Dai, Yifei Li, Deli Chen, Yu Wu, and Zhifang Sui. "Math-shepherd: Verify and reinforce llms step-by-step without human annotations." In Proceedings of the 62nd Annual Meeting of the Association for Computational Linguistics (Volume 1: Long Papers), pp. 9426-9439. 2024.
>
> ---
>
> > **Question:**
> In section 6.2, the authors state, "(clip fraction) reduction becomes more pronounced as the proportion of future information increases. This suggests that future-aware training stabilizes policy updates" (lines 405-407). However, on lines 417-419, they state, "This highlights that excessive incorporation of future information may lead to training instability." Both experiments were conducted using the same settings (MPO-10%/20%/30%/40%) but appear to have contradictory conclusions. I hope the authors can clarify this for me.
>
> ---
>
> **Response to Question:**
>
> Thank you for your careful review. In fact, these two experimental results are not contradictory. Incorporating more future information significantly reduces the variance of the ratios during training, resulting in less frequent gradient clipping (as shown in Figure 4).
>
> However, a lower gradient clipping rate does not necessarily lead to better final optimization results. When the proportion of future information becomes too large, the approximation strategies may introduce additional bias, which can diminish the optimization effect. Therefore, there **exists an optimal proportion** of future information introduced (10–20%), which both stabilizes the training process (i.e., reduces gradient clipping) and achieves the best overall optimization results.
>
> Thank you for your comments. We will clarify this conclusion in the experimental analysis section of the paper.

---

### Official Review · Reviewer_xY8Y · 2025-10-26

**Soundness:** 3
**Presentation:** 2
**Contribution:** 2
**Rating:** 4
**Confidence:** 3

**Summary:**

This paper introduces Multi-Token Policy Gradient Optimization (MPO), a framework that extends token-level policy gradient methods by treating blocks of $K$ future tokens as atomic actions. The key motivations are: (1) to stabilize advantage estimation by aggregating future knowledge through multi-token prediction, and (2) to encourage higher-level planning over coherent text segments rather than individual tokens.
To realize this, MPO replaces the standard per-token importance sampling ratio with a block-level ratio computed over $K$ consecutive tokens. Because directly multiplying per-token ratios can cause high variance, the authors approximate this product using a weighted log-sum of log-ratios, inspired by the Log-COP-TD method.
Empirically, MPO consistently outperforms PPO and GRPO baselines on mathematical reasoning benchmarks (GSM8K and MATH), demonstrating improved stability, lower clip fraction, and higher accuracy in sequence-level reasoning tasks.

**Strengths:**

MPO outperforms GRPO and PPO on mathematical reasoning benchmarks (GSM8K and MATH)

**Weaknesses:**

Although this paper is practical, it lacks sufficient experiments.
- The experimental results are based on relatively small language models (1B or 1.5B).
- The GSM8K and MATH benchmarks are not sufficient to demonstrate the performance of LLMs. Additional tasks should be included (at least those used in the GRPO paper).

**Questions:**

I have following questions about this paper:

1. Additional Experiments (related to Weaknesses)

    The experimental results are based on relatively small language models (1B and 1.5B), which may not be sufficient to evaluate the alignment effect. Are there any additional results on larger models? In addition, the GSM8K and MATH benchmarks may not adequately demonstrate the performance of LLMs. Do the authors have results on other tasks, such as those used in the GRPO paper?

2. About $\beta\_k$

    The authors state that $\beta\_k$ are normalized to $\sum\_{n=1}^K \beta\_n=1$. I assume this means $\beta\_k=\frac{\tilde\beta\_k}{\sum\_{n=1}^K \tilde\beta\_n}$ for arbitrary $\{\tilde\beta\_k\}$. However, the paper seems to adopt a different kind of normalization. Could the authors clarify what “normalization” specifically means in this context?

3. Approximation of $\widetilde{R}$

    Equations (11) and (12) are equivalent when $\beta\_n=1$ for all $n$. However, enforcing $\sum\_{n=1}^K \beta\_n=1$ may introduce additional bias, even if it helps reduce variance, as authors mention in line 289 - 290. How do the authors justify this design choice? Moreover, given the potential bias, is the comparison in Figure 4 fair?

4. Trade-off between variance and bias

    In this paper, both the choice of $K$ and the set of $\{\beta\_n\}$ involve a trade-off between variance and bias. Could the authors clarify how these trade-offs are characterized or managed in the proposed method?

**Details Of Ethics Concerns:**

no concern

---

> ### Author Response · Authors · 2025-11-26
> **Response to Reviewer xY8Y Part 1/3**
>
> > **Weakness 1:** The experimental results are based on relatively small language models (1B or 1.5B).
>
> ---
>
> **Response to Weakness 1:**
>
> MPO does not depend on any specific model architecture and is implemented with a pluggable MTP module, making it directly applicable to language models of different architectures and scales. In our experiments, we seamlessly applied MPO to models with two different architectures—Qwen and LLaMA—and achieved better performance compared to PPO and GRPO. Therefore, our experimental results on the 1B and 1.5B models provide evidence for the effectiveness of MPO to some extent.
>
> Additionally, following your suggestion, we conducted MPO experiments on Deepseek-Qwen2.5-7B to verify its adaptability on a scaled-up model. Due to hardware limitations, we carried out experiments with K=3, and the results are as follows:
>
> | Method | GSM8K | MATH |
> |--------|--------------------------|------------------------------------|
> | Initial | 0.767 | 0.551 |
> | GRPO (Group size=4) | 0.936 | 0.824 |
> | PPO | 0.936 | 0.834 |
> | **MPO** ( Ours, K=3) | **0.941** | **0.851** |
>
> We observe that MPO also outperforms PPO and GRPO on the 7B-scale model, further confirming the effectiveness of our approach. Thank you for your suggestion; we will include these results in the revised version of the paper.
>
> ---
>
> > **Weakness 2:** The GSM8K and MATH benchmarks are not sufficient to demonstrate the performance of LLMs. Additional tasks should be included (at least those used in the GRPO paper).
>
> ---
>
> **Response to Weakness2:**
>
> As in previous works [1,2], we conducted evaluations on two widely used mathematical tasks of different difficulty levels, GSM8K and MATH, comparing our proposed MPO method with the baseline PPO algorithm and the widely used GRPO algorithm. These empirical results can, to some extent, demonstrate the effectiveness of our approach.
>
> Additionally, following your suggestion, we conducted experiments on the HumanEval code dataset to the best of our ability. Based on the MTP model warmed up on mathematical reasoning tasks, we directly used the MBPP dataset as the training set (consisting of 1,399 training samples) and performed 20 epochs of MPO training. The baseline methods were also reproduced under the same experimental settings, and the results are shown in the table below:
>
> | Model| zero-shot | GRPO |PPO | MPO |
> |--------|--------------------------|------------------------------------|-------|-------|
> | Deepseek-qwen-1.5b | 0.451 |0.591| 0.598 | **0.640** (↑4.2%) |
> | llama3.2-1b | 0.354 |0.372| 0.390 | **0.403** (↑1.3%）|
>
> We have indicated the performance improvement of MPO relative to PPO. Based on these results, we observe significant performance improvements for our proposed MPO over both PPO and GRPO on code tasks and on both models. This further demonstrates the effectiveness of our method.
>
> Thank you for your comments. We will include these results in the paper.
>
> ---
>
> [1] Kazemnejad, Amirhossein, Milad Aghajohari, Eva Portelance, Alessandro Sordoni, Siva Reddy, Aaron Courville, and Nicolas Le Roux. "VinePPO: Refining Credit Assignment in RL Training of LLMs." In Forty-second International Conference on Machine Learning. 2024.
>
> [2] Wang, Peiyi, Lei Li, Zhihong Shao, Runxin Xu, Damai Dai, Yifei Li, Deli Chen, Yu Wu, and Zhifang Sui. "Math-shepherd: Verify and reinforce llms step-by-step without human annotations." In Proceedings of the 62nd Annual Meeting of the Association for Computational Linguistics (Volume 1: Long Papers), pp. 9426-9439. 2024.
>
> ---
>
> > **Question 1** (related to Weaknesses)
> The experimental results are based on relatively small language models (1B and 1.5B), which may not be sufficient to evaluate the alignment effect. Are there any additional results on larger models? In addition, the GSM8K and MATH benchmarks may not adequately demonstrate the performance of LLMs. Do the authors have results on other tasks, such as those used in the GRPO paper?
>
> ---
>
> **Response to Question 1:**
>
> Please refer to the additional experiments we have conducted in response to Weakness 1 and Weakness 2 as you listed. We hope the above results address your concerns, and thank you for raising this question!

---

> ### Author Response · Authors · 2025-11-26
> **Response to Reviewer xY8Y Part 2/3**
>
> > **Question 2:** About $ \beta_k $. The authors state that $ \beta_k $ are normalized to $ \sum_{n=1}^{K} \beta_n = 1 $. I assume this means  $\frac{\tilde{\beta}_k}{\sum_n \tilde{\beta}_n}$ for arbitrary $ \tilde{\beta}_k $. However, the paper seems to adopt a different kind of normalization. Could the authors clarify what “normalization” specifically means in this context?
>
> ---
>
> **Response to Question 2:**
>
> Our explanation of the MTP weight setting in the paper was somewhat unclear; we will revised the relevant sections in the text and would like to clarify the process here.
>
> Actually, when setting the MTP weights, two hyperparameters are involved: the weight for the first MTP module (responsible for next-2 token prediction), denoted as \beta_2, and the decay factor, denoted as \lambda. After manually setting the above parameters, the values of \beta_3 through \beta_K are determined by the following formula:
>
> $$
> \beta_k = \beta_2 \times \lambda^{k-2}
> $$
>
> Then,
>
> $$
> \beta_1 = 1 - \sum_{k=2}^K \beta_k
> $$
>
> so that the total sum is guaranteed to be 1. No additional normalization is applied in this process.
>
> We will add the above description to the revised version of the paper to explain the weight settings. Thank you for your careful comment.
>
> ---
>
> > **Question 3:**
> Approximation of $\tilde{R}$. Equations (11) and (12) are equivalent when $ \beta_n = 1 $ for all $ n $.  However, enforcing $\sum_{n=1}^{K} \beta_n = 1$ may introduce additional bias,  even if it helps reduce variance, as authors mention in line 289–290.  How do the authors justify this design choice?  Moreover, given the potential bias, is the comparison in Figure 4 fair?
>
> ---
>
> **Response to Question 3:**
>
> Theoretically, aggregating K steps in the policy gradient optimization process is a means to **reduce bias**. To illustrate this, we have added a qualitative analysis of the effect of K on the overall advantage estimation bias in Appendix D. From a theoretical perspective, increasing the length of the action interval K considered during optimization can effectively reduce the bias.
>
> On this basis, the approximation of R is a **"further trade-off"** made to stabilize training, and the overall potential bias should be less than that introduced by the approximation of R. If no approximation algorithm is used and Equation (11) is applied directly in training, the variance introduced by observing multi-step actions to reduce bias can cause the training to collapse according to our experiments. Therefore, the bias in MPO is controllable, structurally reasonable, and is used to ensure the stability of the training process.
>
> From this perspective, to balance the variance introduced by fusing multi-step decision ratios, the probability approximation process in MPO, like the KL regularization in PPO and the relative in-group advantage estimation in GRPO, inevitably introduces bias [1,2]. In other words, the comparison in Figure 4 is fair to a reasonable extent.
>
> ---
>
> [1] Schulman, John, Filip Wolski, Prafulla Dhariwal, Alec Radford, and Oleg Klimov. "Proximal policy optimization algorithms." arXiv preprint arXiv:1707.06347 (2017).
>
> [2] Shao, Zhihong, Peiyi Wang, Qihao Zhu, Runxin Xu, Junxiao Song, Xiao Bi, Haowei Zhang et al. "Deepseekmath: Pushing the limits of mathematical reasoning in open language models." arXiv preprint arXiv:2402.03300 (2024).

---

> ### Author Response · Authors · 2025-11-26
> **Response to Reviewer xY8Y Part 3/3**
>
> > **Question 4:**
> In this paper, both the choice of K and the set of $\beta_n$ involve a trade-off between variance and bias. Could the authors clarify how these trade-offs are characterized or managed in the proposed method?
>
> ---
>
> **Response to Question 4:**
>
> K (the number of future tokens) and $\beta_n$ (the weights of each prediction head) indeed jointly determine the bias–variance trade-off. Our approach controls this trade-off through structured design rather than arbitrary hyperparameter tuning.
>
> ---
>
> ### The Role of K:
>
> Although the above analysis shows that, in theory, optimizing strategies that consider K-step decisions is a bias-reducing but variance-increasing trade-off, in MPO experiments, increasing K leads to a significant increase in variance, making the overall training process highly unstable. Therefore, in our implementation, we additionally use an approximate ratio calculation to stabilize training as a further trade-off.
>
> Keeping the total weights of MTP ratio at 20%, we conduct experiments with Deepseek-Distill-Qwen2.5-1.5b to analyse the effect of K (also shown in Table 4 in Appendix B.2):
>
> | Model| K=2 | K=3 | K=5 |
> |--------|--------------------------|------------------------------------|-------|
> | GSM8K | 0.871 |0.875| **0.882** |
> | MATH | 0.771 |0.753| **0.779** |
>
> As shown in the Table, based on current experimental results, increasing K from 3 to 5 leads to a change in performance. We believe that continuing to increase K may still offer potential benefits, but may also introduce additional bias that could affect training performance. In future experiments, we will further explore low-overhead implementations of MTP, making it more efficient to investigate the effect of policy aggregation for K > 5 scenarios.
>
> ---
>
> ### The Role of $\beta_n$ :
>
> $\beta_{n}$ controls **the contribution of deeper prediction heads**. For the MTP module, we adopt a gradually decaying strategy to allocate the weights, ensuring that deeper predictions (which are noisier) have a lower proportion. Due to the approximate calculation of $\tilde{R}$, increasing $\beta_n$'s contribution to variance stabilization introduces additional bias.
>
> Let $\beta_{mtp}=\sum_{n=2}^K \beta_n$, which represents the sum of weights of the MTP module. Experiments in Figures 4 and 5 demonstrate that increasing $\beta_{mtp}$ effectively reduces training variance. When $\beta_{mtp}$ lies within the 0–20% range, this effectively brings performance gains to MPO. However, as $\beta_{mtp}$ continues to increase, although variance is further reduced, the potential bias resulting from the trade-off hinders further performance improvement.
>
> Thank you again for your careful review and comments, we hope this clarification addresses your concerns.

---

### Official Review · Reviewer_14o5 · 2025-10-31

**Soundness:** 2
**Presentation:** 3
**Contribution:** 2
**Rating:** 4
**Confidence:** 3

**Summary:**

This paper proposes **Multi-Token Policy Gradient Optimization (MPO)**, a novel framework for post-training large language models (LLMs) using reinforcement learning. The central claim is that standard policy-gradient methods like PPO operate at a token-level granularity, which is a poor match for sequence-level rewards in complex reasoning tasks like mathematics. To address this, MPO redefines the policy gradient "action" as a **block of $K$ future tokens**. It computes the importance sampling ratio over this entire multi-token block rather than a single token.

To implement this, the method incorporates auxiliary **Multi-Token Prediction (MTP) modules**, which are first "warmed up" to predict future token probabilities. To manage the high variance that arises from multiplying multiple probability ratios, MPO introduces a **log-space approximation** (a weighted log-sum) to stabilize training. Experiments conducted on the GSM8K and MATH benchmarks show that MPO consistently outperforms token-level baselines, PPO and GRPO, in mathematical reasoning accuracy.

**Strengths:**

-  **Important Problem:** The paper addresses a significant and timely problem: the granularity mismatch between the token-level optimization of standard policy gradient methods and the sequence-level, holistic nature of rewards in complex reasoning tasks. Improving RL alignment for multi-step reasoning is a critical research direction.

- **Sufficient Literature Review:** The paper provides a comprehensive review of related work. It clearly positions MPO relative to both existing policy gradient methods (PPO, GRPO, DAPO) and the separate field of multi-token prediction (MTP) techniques.

- **Novel Method:** The core contribution—modifying the policy gradient objective to compute importance sampling ratios over multi-token blocks—is a novel and interesting approach for LLM alignment.

- **Clarity and Presentation:** The paper is generally well-written and easy to follow. The method is explained clearly, and the inclusion of diagrams like Figure 1 and Figure 2 is helpful for understanding the MTP module's role and the overall MPO training process.

**Weaknesses:**

- **Insufficient Motivation:** The paper is not adequately motivated. While it claims MPO "better captures the structure of reasoning", it doesn't provide a strong intuitive or theoretical explanation for *why* computing the importance ratio over $K$ future tokens leads to a more effective or stable policy update compared to the standard single-token ratio.



- **Methodological Soundness and Cost:** The proposed method's reliance on *new* MTP modules raises significant concerns about soundness and efficiency:

  - **Redundancy:** The authors do not explain why the original, backbone LLM cannot be used to auto-regressively compute the probabilities for the $K$ future tokens. The introduction of separate MTP modules seems redundant and adds significant complexity.

  - **Parameter Overhead:** These new modules add a substantial number of parameters, increasing the model size significantly (e.g., 1.52x to 2.05x for $K=4$ on a Llama3.2-1B model).

  - **Training Complexity:** The MTP modules require a separate "warm-up" phase before MPO training can begin, complicating the overall training pipeline.

  - **Computational Cost:** When computing the MPO loss, the objective (Eq. 12) appears to require $K$ forward passes (or at least $K$ probability lookups) *for each token* $t$ in the trajectory to calculate the multi-token ratio. This represents a potential $K$-fold increase in computational cost for the loss computation, which is a major drawback.

- **Limited Empirical Evaluation:** The empirical results could be improved. The method is only tested on two math benchmarks (GSM8K, MATH) and compared against only two baselines (PPO, GRPO). A key omission is a comparison against **DAPO**, which is cited as an advanced method for reasoning tasks and would serve as a much stronger baseline.

- **Insignificant Performance vs. Cost:** The reported performance gains are not very significant, especially when weighed against the massive increase in computational and parameter costs. For example, on GSM8K (Deepseek-Qwen2.5-1.5B), MPO ($K=5$) improves over PPO by only 1.6 percentage points (0.882 vs. 0.866), while on MATH the gain is also 1.6 points (0.779 vs. 0.763). This marginal improvement does not seem to justify the 26-48% slowdown in training iteration speed and the 1.2x-2.05x increase in model size.

**Questions:**

1. Could the authors clarify the precise reason for introducing separate MTP modules? Why not use the backbone LLM's own decoder to compute the future token probabilities $\pi_{\theta}(o_{i,t+n}|o_{i,1:t+n-1})$ for the importance ratio? What advantages do the specialized MTP module architectures (as shown in Figure 1b)  provide over the standard backbone?

2. The paper motivates MPO as "encouraging higher-level planning" Could the authors elaborate on this mechanism? How does aggregating $K$ ratios via a weighted log-sum (Eq. 12) translate to a better policy gradient for long-term reasoning, beyond the variance reduction shown in Figure 4a?

3. The best results are reported for $K=5$. However, the cost analysis in Table 3 only provides data for $K=2$ and $K=4$. What are the specific model size and training speed costs for the $K=5$ setting, which yielded the best performance?

4. Given that DAPO is also a policy gradient method designed to improve upon PPO/GRPO for reasoning tasks, why was it omitted from the experimental comparison? How would the authors hypothesize MPO compares to DAPO in terms of both performance and computational efficiency?

---

> ### Author Response · Authors · 2025-11-26
> **Response to Reviewer 14o5 Part 1/4**
>
> > **Weakness 1:**
>  Insufficient Motivation: The paper is not adequately motivated. While it claims MPO "better captures the structure of reasoning", it doesn't provide a strong intuitive or theoretical explanation for why computing the importance ratio over K future tokens leads to a more effective or stable policy update compared to the standard single-token ratio.
>
> > **Question 2:**
> The paper motivates MPO as "encouraging higher-level planning" Could the authors elaborate on this mechanism? How does aggregating K ratios via a weighted log-sum (Eq. 12) translate to a better policy gradient for long-term reasoning, beyond the variance reduction shown in Figure 4a?
>
> ---
>
> **Response to Weakness 1 & Question 2:**
>
> We have added Figure 7 in Appendix A of the revised paper to illustrate our motivation more clearly.
>
> In reasoning tasks, key decision points (such as the “a” in “a=5” or the “b” in “b=a^2” as shown in Figure 7) often influence the generation of multiple subsequent tokens. Overly decomposed optimization as in NTP can lead the model to favor myopic or shortcut behaviors, which is detrimental to learning global structures and long-term dependencies.
>
> As shown in Figure 7, MPO attempts to capture the multi-step impact of decisions by aggregating ratios over the next K tokens (Equations 9 and 12). Compared to single-step ratios, MPO can more accurately reflect **“the effect of a local decision on a more complete reasoning action.”** At key points that determine the subsequent reasoning substructure, MPO enables the model to estimate the advantage of the current action from the perspective of more complete formula or code-level semantic actions by aggregating K-step ratios, thereby enhancing the optimization performance of the algorithm.
>
> Thank you for your suggestion. We will add the above explanations to the paper to further refine and update the motivation section.
>
> ---
>
> > **Weakness 2.1** Redundancy: The authors do not explain why the original, backbone LLM cannot be used to auto-regressively compute the probabilities for the K future tokens. The introduction of separate MTP modules seems redundant and adds significant complexity.
>
> > **Question 1**:
> Could the authors clarify the precise reason for introducing separate MTP modules? Why not use the backbone LLM's own decoder to compute the future token probabilities K for the importance ratio? What advantages do the specialized MTP module architectures (as shown in Figure 1b) provide over the standard backbone?
>
> ---
>
> **Response to Weakness 2.1 and Question 1:**
>
> Regarding the redundancy issue, on the one hand, directly aggregating the ratios of K tokens based on the auto-regressive model itself does not allow for gradient back-propagation across sequence positions, which is **incompatible with the optimization process designed by MPO**.
>
> On the other hand, even if hidden representations are repeatedly fed back as inputs to enable gradient flow, involving the backbone model in the prediction of future tokens during training may interfere with its inherent next-token prediction patterns, whereas the MTP module can be **flexibly detached after training**.
>
> Moreover, MPO only modifies the advantage estimation method in the RLHF process and does not restrict the implementation of multi-token prediction strategies. In summary, the proposed MPO is already one of the optimal solutions for aggregating multi-token decision making.
>
> Thank you for your comments.

---

> ### Author Response · Authors · 2025-11-26
> **Response to Reviewer 14o5 Part 2/4**
>
> > **Weakness 2.2**
> Parameter Overhead: These new modules add a substantial number of parameters, increasing the model size significantly (e.g., 1.52x to 2.05x for K=4 on a Llama3.2-1B model).
>
> ---
>
> **Response to Weakness 2.2:**
>
> The MTP implementation adopted by MPO is a pluggable module, and the additional parameters it introduces do not participate in the inference or testing phase, ensuring a fair comparison with other baselines.
>
> At the same time, the additional parameters do not significantly influence the efficiency of the method. The table below provides a comparison of the average overhead and performance of PPO, GRPO, and our proposed MPO, all based on Deepseek-Qwen2.5-1.5B under identical hardware and environment settings:
>
> | Method | Avg Memory Usage | Avg Training Time (per Step) | GSM8K (Acc.)|MATH (Acc.)|
> |--------|--------------------------|------------------------------------|-------|------|
> | PPO | 22.1% | 25.6 s | 0.866 |0.761|
> | GRPO (Group size=4) | 30.6% | 43.7 s | 0.850 |0.751|
> | **MPO** (Ours, K=5) | 31.0% | 33.1 s | **0.882** |**0.779**|
>
> According to the table, we can see that our proposed MPO, even when using ( K=5 ), achieves memory usage comparable (with difference only 0.4%) to the widely recognized GRPO, while offering a significant advantage in time efficiency (30% faster).
>
> Thank you for your comment. We will include the above comparison results in the revised version of our paper.
>
> ---
>
> > **Weakness 2.3**
> Training Complexity: The MTP modules require a separate "warm-up" phase before MPO training can begin, complicating the overall training pipeline.
>
> ---
>
> **Response to Weakness 2.3:**
>
> Following prior works [1,2], our method also includes an MTP warm-up stage before MPO training. This phase is necessary to stabilize the newly introduced MTP modules and ensure they fit with the policy model and provide reliable future token prediction.
>
> In practice, the warm-up is **lightweight** (only 1 epoch, 12 minutes) compared to the main MPO phase (6.5h~7h on MATH) and follows the same procedure as existing approaches. Therefore, while warm-up introduces an additional step, it is a standard and essential component of MTP-based methods rather than extra complexity unique to our approach.
>
> We appreciate the reviewer’s suggestion.
>
> ---
>
> [1]Cai, Tianle, Yuhong Li, Zhengyang Geng, Hongwu Peng, Jason D. Lee, Deming Chen, and Tri Dao. "Medusa: Simple LLM Inference Acceleration Framework with Multiple Decoding Heads." In International Conference on Machine Learning, pp. 5209-5235. PMLR, 2024.
>
> [2]Ankner, Zachary, Rishab Parthasarathy, Aniruddha Nrusimha, Christopher Rinard, Jonathan Ragan-Kelley, and William Brandon. "Hydra: Sequentially-Dependent Draft Heads for Medusa Decoding." In First Conference on Language Modeling.
>
> ---
>
> > **Weakness 2.4**
> Computational Cost: When computing the MPO loss, the objective (Eq. 12) appears to require K forward passes (or at least K probability lookups) for each token t in the trajectory to calculate the multi-token ratio. This represents a potential k-fold increase in computational cost for the loss computation, which is a major drawback.
>
> ---
>
> **Response to Weakness 2.4:**
>
> Your understanding of our loss computation process is correct. But in fact, the overall computational cost **does not increase k-fold**. This is because all logits needed for multi-token ratios are obtained from a single standard forward pass; the MTP module generates predictions for K future tokens in parallel within one forward pass, without requiring K separate model evaluations.
>
> In other words, K is simply an additional head dimension, not an extra model pass dimension. Furthermore, as shown in our experimental results in our response to Weakness 2.2, our method **does not incur K times the memory or time overhead**. On the contrary, our method has lower time overhead and better performance than GRPO.
>
> We hope this clarifies your misunderstanding, and thank you for your comment.

---

> ### Author Response · Authors · 2025-11-26
> **Response to Reviewer 14o5 Part 3/4**
>
> > **Weakness 3:**
> Limited Empirical Evaluation: The empirical results could be improved. The method is only tested on two math benchmarks (GSM8K, MATH) and compared against only two baselines (PPO, GRPO). A key omission is a comparison against DAPO, which is cited as an advanced method for reasoning tasks and would serve as a much stronger baseline.
>
> > **Question 4:**
> Given that DAPO is also a policy gradient method designed to improve upon PPO/GRPO for reasoning tasks, why was it omitted from the experimental comparison? How would the authors hypothesize MPO compares to DAPO in terms of both performance and computational efficiency?
>
> ---
>
> **Response to Weakness 3 and Question 4:**
>
> Following previous works [1,2], we conduct evaluations on two widely used mathematical problem sets—GSM8K and MATH, which cover different levels of difficulty—and compare our proposed MPO method with the baseline PPO algorithm as well as the widely used GRPO algorithm. These empirical results can, to some extent, demonstrate the effectiveness of our approach.
>
> Additionally, following your suggestion, we conducted experiments on the HumanEval code dataset to the best of our ability. Based on the MTP model trained for mathematical reasoning tasks, we directly used the MBPP dataset as the training set (a total of 1,399 training examples) and performed 20 epochs of MPO training. Under the same experimental settings, we also re-implemented the baseline methods. The results are shown in the following table:
>
> | Model| zero-shot | GRPO |PPO | MPO |
> |--------|--------------------------|------------------------------------|-------|-------|
> | Deepseek-qwen-1.5b | 0.451 |0.591| 0.598 | **0.640** (↑4.2%) |
> | llama3.2-1b | 0.354 |0.372| 0.390 | **0.403** (↑1.3%) |
>
> We have indicated the performance improvement of MPO over PPO. Based on these results, we observe that our proposed MPO achieves significant performance gains over PPO and GRPO in code tasks on both models. This further demonstrates the effectiveness of our method.
>
> Furthermore, we have reproduced DAPO and reported the DAPO baseline results (with trajectory group size 4) on the two datasets. The table below highlights the performance improvement of MPO over DAPO:
>
> | Result on GSM8K| zero-shot |PPO | GRPO| DAPO | MPO|
> |--------|--------------------------|------------------------------------|-------|-------|-------|
> | Deepseek-qwen-1.5b | 0.716 | 0.866 | 0.850 | 0.862 | **0.882**  (↑ 2%)|
> | llama3.2-1b | 0.311 | 0.614 | 0.553 | 0.618 | **0.632** (↑ 1.4%)|
>
> | Result on MATH| zero-shot |PPO | GRPO| DAPO | MPO|
> |--------|--------------------------|------------------------------------|-------|-------|-------|
> | Deepseek-qwen-1.5b |0.495 | 0.761 | 0.751 | 0.769 | **0.779** (↑ 1%) |
> | llama3.2-1b | 0.150 | 0.337 | 0.330 | 0.355 | **0.360** (↑ 0.5%)|
>
> The above results show that our proposed MPO still outperforms DAPO on GSM8K and MATH, further confirming the validity of our method.
>
> We will include these results in the updated version of our paper. Thank you for your comments.
>
> ---
>
> [1] Kazemnejad, Amirhossein, Milad Aghajohari, Eva Portelance, Alessandro Sordoni, Siva Reddy, Aaron Courville, and Nicolas Le Roux. "VinePPO: Refining Credit Assignment in RL Training of LLMs." In Forty-second International Conference on Machine Learning. 2024.
>
> [2] Wang, Peiyi, Lei Li, Zhihong Shao, Runxin Xu, Damai Dai, Yifei Li, Deli Chen, Yu Wu, and Zhifang Sui. "Math-shepherd: Verify and reinforce llms step-by-step without human annotations." In Proceedings of the 62nd Annual Meeting of the Association for Computational Linguistics (Volume 1: Long Papers), pp. 9426-9439. 2024.

---

> ### Author Response · Authors · 2025-11-26
> **Response to Reviewer 14o5 Part 4/4**
>
> > **Weakness 4:**
> Insignificant Performance vs. Cost: The reported performance gains are not very significant, especially when weighed against the massive increase in computational and parameter costs. For example, on GSM8K (Deepseek-Qwen2.5-1.5B), MPO (K=5) improves over PPO by only 1.6 percentage points (0.882 vs. 0.866), while on MATH the gain is also 1.6 points (0.779 vs. 0.763). This marginal improvement does not seem to justify the 26-48% slowdown in training iteration speed and the 1.2x-2.05x increase in model size.
>
> ---
>
> **Response to Weakness 4:**
>
> Please refer to our response to Weakness 2.2, we provided a table comparing the computational overhead. Compared with the widely recognized GRPO algorithm, MPO has comparable memory consumption, significantly better time efficiency, and superior performance. This already demonstrates the effectiveness of our method.
>
> ---
>
> > **Question 3:**
> The best results are reported for K=5. However, the cost analysis in Table 3 only provides data for K=2 and K=4. What are the specific model size and training speed costs for the  setting, which yielded the best performance?
>
> ---
>
> **Response to Question 3:**
>
> We sincerely apologize; due to revisions in the definition of K during different iterations of the paper, Table 3 in the manuscript incorrectly labels the values of K (the correct values should be (K=3) and (K=5)).
>
> Thank you for your careful review, we will correct this labeling in the updated version of the paper.

---

### Official Review · Reviewer_7sKG · 2025-11-01

**Soundness:** 3
**Presentation:** 3
**Contribution:** 3
**Rating:** 6
**Confidence:** 4

**Summary:**

This paper proposes Multi-Token Policy Gradient Optimization (MPO) that extends token-level policy optimization to multi-token policy optimization for matching the sequence-level nature of rewards in reasoning tasks. More specifically, MPO calculates importance sampling ratios (of PPO) over K future tokens. Then, MPO proposes the MPO objective in Eq. 13. This paper evaluates MPO on two representative mathematical reasoning benchmarks including GSM8K and MATH by using two language models including DeepSeek-Distilled-Qwen2.5-1.5B and Llama3.2-1B.

**Strengths:**

- S1. [Presentation] First of all, this paper is very well written and organized.

- S2. [Motivation] It is clearly motivated why multi-token policy optimization of LLMs is required for mathematical reasoning.

- S3. [Analysis] Besides the main results, this paper provides a comprehensive empirical analysis on the effectiveness of MPO. It includes (1) variance of importance sampling ratios, (2) effect of future information injection, (3) effect of MTP hyper-parameters, and (4) reliability of incorporated future information. Also, this paper provides an analysis of time and memory cost, comparing MPO with PPO.

**Weaknesses:**

- W1. [Performance] One of main weaknesses of this paper may be the performance gain, compared to the training cost. According to Figure 3, in case of DeepSeek-Qwen2.5-1.5B, MPO achieves the accuracy of 0.882, while PPO provides 0.866. According to Table 3, MPO takes 30% more training time and 40% more memory.

**Questions:**

- Q1. This paper uses MPO to train relatively small language models such as Deepseek-Qwen2.5-1.5B and Llama3.2-1B. Is the performance improvements increased, if larger language models are used?

- Q2. Current implementation of MPO is based on PPO. Is it possible to apply MTP to GRPO?

---

> ### Author Response · Authors · 2025-11-26
> **Response to Reviewer 7sKG**
>
> > **W1.** [Performance] One of main weaknesses of this paper may be the performance gain, compared to the training cost. According to Figure 3, in case of DeepSeek-Qwen2.5-1.5B, MPO achieves the accuracy of 0.882, while PPO provides 0.866. According to Table 3, MPO takes 30% more training time and 40% more memory.
>
> ---
>
> **Response to W1:**
>
> In general, reinforcement learning tends to introduce additional computational overhead. The table below shows a comparison of the average overhead and performance of PPO, GRPO, and our proposed MPO methods, all based on Deepseek-Qwen2.5-1.5B under identical hardware and environment configurations:
>
> | Method | Avg Memory Usage | Avg Training Time (per Step) | GSM8K (Acc.)|MATH (Acc.)|
> |--------|--------------------------|-----------------------------|-------|------|
> | PPO | 22.1% | 25.6 s | 0.866 |0.761|
> | GRPO (Group size=4) | 30.6% | 43.7 s | 0.850 |0.751|
> | **MPO** (Ours, K=5) | 31.0% | 33.1 s | **0.882** |**0.779**|
>
> Compared with the widely recognized GRPO algorithm, our proposed MPO exhibits **equivalent memory consumption**, significantly **better time efficiency**, and a **2.8%~3% performance improvement**.
>
> Specifically, when the trajectory group size is set to 4, the average per step time overhead of GRPO is 43.7 seconds, which is about 30% higher than our MPO with (K=5). The key advantage of MPO lies in the fact that it does not require multiple trajectories for within-group advantage estimation, so the memory consumption per sample is also on par with GRPO.
>
> Thank you for your comment. We will add the above comparison results to the revised version of our paper.
>
> ---
>
> > **Q1.** This paper uses MPO to train relatively small language models such as Deepseek-Qwen2.5-1.5B and Llama3.2-1B. Is the performance improvements increased, if larger language models are used?
>
> ---
>
> **Response to Q1:**
>
> To capture the multi-step influence of decision-making during the optimization process, MPO acquires and aggregates block-level ratios with the help of pluggable MTP modules, which **is not dependent on any specific model architecture**. MPO can be directly applied to language models of varying architectures and scales.
>
> In our experiments, we seamlessly applied MPO to both Qwen and LLaMA models, which differ in architecture, and achieved better performance compared to PPO and GRPO. Thus, the experimental results on 1B and 1.5B models provide a certain degree of validation for the effectiveness of MPO.
>
> Additionally, following your suggestion, we conducted MPO experiments based on Deepseek-Qwen2.5-7B to evaluate its adaptability on scaled models. Due to hardware limitations, we conducted MPO experiments with K=3, and the results are as follows:
>
> | Method | GSM8K | MATH |
> |--------|--------------------------|------------------------------------|
> | Initial | 0.767 | 0.551 |
> | GRPO (Group size=4) | 0.936 | 0.824 |
> | PPO | 0.936 | 0.834 |
> | **MPO** (Ours, K=3) | **0.941** | **0.851** |
>
> We observe that MPO also outperforms PPO and GRPO on models at the 7B scale, which further confirms the effectiveness of our method.
>
> Thank you for your suggestion; we will include these results in the revised version of our paper.
>
> ---
>
> > **Q2.** Current implementation of MPO is based on PPO. Is it possible to apply MTP to GRPO?
>
> ---
>
> **Response to Q2:**
>
> According to Equation (13) and its explanation in our paper, our objective function can be transformed into the form of GRPO. MPO only balances the training process by adjusting the importance sampling parameters, and in theory, it can be adapted to GRPO. However, implementing MPO based on GRPO may encounter other challenges, such as：
>
> - the interaction between GRPO’s group-based reward estimation and the variance introduced by the K-token joint ratio, which may require re-balancing the bias–variance trade-off in its advantage estimation;
> - the increased computational overhead of the MTP module when multiple trajectories are involved.
>
> Meanwhile, we are attempting to incorporate MPO into GRPO for further validation, and we will make every effort to provide updated results before the deadline.
>
> Thank you for your suggestion.

---

### Author Response · Authors · 2025-12-03
**Summary of Rebuttal phase**

Dear Reviewers/AC/SAC:

We sincerely thank you for taking the time to review our paper.

During the rebuttal phase, we try our best to address the reviewers’ concerns with experiments and clarifications, and thoroughly revised our paper based on all the suggestions and new results. For ease of review, we summarize the acknowledged contributions of our paper, and our efforts to address the raised concerns.

---

## Acknowledged Contributions

- **Research significance:**   We **extend token-level policy optimization to multi-token policy optimization** (Reviewer 7sKG) to **address the important challenge of the granularity mismatch** (Reviewer fG3C) between token-level optimization and sequence-level reward signals, which is **a significant and timely problem** (Reviewer 14o5).

- **Methodology:** We propose Multi‑Token Policy Gradient Optimization (MPO) for **matching the sequence-level nature of rewards in reasoning tasks** (Reviewer 7sKG), which is a **novel** (Reviewer 14o5, fG3C), **practical** (Reviewer xY8Y), **interesting** (Reviewer 14o5) and **intuitively sound** (Reviewer fG3C) approach for LLM alignment.

- **Empirical analysis:** MPO **outperforms GRPO and PPO on mathematical reasoning benchmarks** (Reviewer xY8Y). Besides the main results, this paper provides a **comprehensive empirical analysis** (Reviewer 7sKG) on the effectiveness of MPO .

We are also pleased that reviewers 7sKG, 14o5, and fG3C gave positive comments “**well-written**” on the presentation of our paper.

---

## Major Concerns:

We sincerely appreciate the reviewers’ valuable comments and have made every effort to provide thorough responses.

> **Concerns about the adequacy of the motivation statement:** While reviewer 7sKG  considered our motivation sufficient, reviewers 14o5 and fG3C  suggested that we further strengthen and clarify the theoretical and intuitive justification for our proposed method.

- We **provide new illustrative figure and examples** to intuitively demonstrate the block‑structured semantic patterns observed in reasoning processes and the necessity of extending beyond token‑level optimization.

- We **provide mathematical derivations** to theoretically prove the upper bound of optimization bias is reduced by observing block-wise actions.

> **Concerns about the generalization of the method:** Though all reviewers acknowledged the performance improvements achieved by our approach, we receive concerns about the method’s generalization ability.

We conducted new experiments from three perspectives:

- **Task level**: We added results on the HumanEval code‑generation benchmark.
- **Scale Model**: We included experiments using the DeepSeek‑Distilled‑Qwen‑7B model.
- **Baseline coverage**: We reproduce the latest DAPO algorithm as a strong baseline.

The proposed MPO **consistently maintained its performance advantage over baselines**, providing strong evidence of its generalization capability.

> **Concerns about the overhead of the method:** Reviewer 7sKG and 14o5 raised concerns about the overhead, noting that MPO may to some extent increase the training cost.

We **conducted and presented experiments** comparing the actual memory overhead and training time between MPO and GRPO, show case that the additional overhead is well controlled.

---

## Other issues:

The following summarizes our efforts to address other specific concerns raised by the reviewer:

- **Reviewer 7sKG** asked whether MPO could be implemented based on GRPO. *We have confirmed the feasibility of GRPO-based MPO and are conducting experiments to further validate its performance.*

- **Reviewer 14o5** concerns about the efficiency and parameter overhead of MTP module. *We carefully analyzed and explained why the current MTP represents one of the optimal strategies, and clarified that the extra parameters do not cause unfair comparisons*

- **Reviewer fG3C** expects an explanation of how the hyperparameters in MPO influence the trade-off between overall variance and bias. *We provided a detailed clarification of how these hyperparameters affect the bias of the optimization strategy based on our experimental results.*

- **Reviewer xY8Y** raises issue about justification for cutting sequence into blocks and the setting of block size K, which also led to other concerns. *We clarified that there is a misunderstanding on the implementation of MPO and explicitly explained our analysis of MPO hyperparameters along with experiments reported in the paper.*

---

Unfortunately, after we submitted our rebuttal, an unexpected technical issue occurred, and we were unable to engage in further discussions. We hope that the summary above helps provide a clearer understanding of our work.

We sincerely thank you for your time and effort in reviewing our paper.

Best regards,

Authors of Submission 10280

---

> ### Author Response · Authors · 2025-12-04
> **Summary of Revision**
>
> Based on the reviewers’ comments during the rebuttal phase, as well as our responses and the additional experiments, we have thoroughly revised the entire paper. The main updates are summarized below:
>
> - **Motivation** (Based on the discussion with Reviewer 14o5, fG3C): We strengthened the motivation section in Section 1 and revised the right panel of Figure 1 to more intuitively illustrate the advantages of multi-token aggregation optimization. The original MTP implementation details have been moved to Appendix A.
>
> - **Bias–variance trade-off** (Based on the discussion with Reviewer xY8Y): Section 4.2 (lines 275–280) has been revised to clarify the introduction of multiple tokens and the approximation strategy in MPO. The full theoretical derivation is now provided in Appendix B.2.
>
> - **New experiments** (Based on the discussion with all reviewers): (i) results of Deepseek-qwen-7b on mathematical reasoning tasks (**Figure 3**); (ii) the DAPO baseline (**Figure 3**); (iii) MPO performance on the HumanEval benchmark (**Table 2**); and (iv) its memory and time costs (**Table 5**). And also, we conduct extra experiments to check the hyperparameter grid search results of MPO. Based on our latest results, we update the MPO performance with Deepseek-qwen-1.5b on MATH with further 1% improvement (**Figure 3, Table3 (b)**).
>
> - **Notation consistency** (Based on the discussion with Reviwer 14o5, xY8Y): We unified the use of $K$ and $\beta_k$ in Q2 and Q4 of Section 6.2 to eliminate potential ambiguities.
>
> The other issues discussed with the reviewers have also been revised accordingly in the relevant sections of the manuscript. In addition, we carefully reviewed the entire text to further enhance its logical coherence and readability.
>
> We sincerely thank all the reviewers for their careful reviews and comments.

---

### Note · Authors · 2026-01-26

I have read and agree with the venue's withdrawal policy on behalf of myself and my co-authors.

---

### Meta-Review · Area_Chair_DtRV · 2025-12-18

**Summary:**

Following the rebuttal, several significant concerns shared by the reviewers remain unresolved:

First, the scope of the experimental evaluation remains insufficient. While the authors incorporated additional results—including a new baseline (DAPO) and a breakdown of computational costs—the empirical evidence is still too narrow to support the paper's core claims. Specifically, the evaluation is limited to only two datasets and a small selection of models. To ensure that the conclusions are both convincing and reliable, the authors must include a broader range of datasets and more baselines.

Second, the underlying motivation of the work requires further clarification. The current assertion that the proposed method "better captures the structure of reasoning" remains largely handwavy and lacks a rigorous theoretical or empirical justification.

**Reviewer Concerns:**

The authors have partially addressed the concerns below:
- Is it possible to apply the method to GRPO? [7sKG]
- Redundant design and complicated training of the separate MTP module. [14o5]
- Computation cost is significantly increased. [14o5]
- Justification of design choices of \beta and \tilde{R}. [xY8Y]
- Lack of Justification for hyperparameter K and limited discussion bias-variance trade-off. [xY8Y, 14o5, fG3C]
- Contradictory experimental results of including more vs. less future information. [fG3C]

However, the following concerns are still outstanding:
- Insufficient motivation. No strong intuitive or theoretical explanation of why the method is good. [14o5, fG3C]
- The performance gain is limited compared to the significantly increasing training time. [7sKG, 14o5]
- How does the method generalize to larger models? [7sKG]
- Limited experiments. Only two benchmarks and two baselines are included and models are fairly small. [7sKG, 14o5, xY8Y, fG3C]

**Reviewer Scores:**

Given full discussion, I do not think reviewers will likely change their scores as their shared major concern on the limited experiments still remain.

---

### Decision · Program_Chairs · 2026-01-26

Reject